# Stochastic simulation of reference rainfall scenarios for hydrological applications using a universal multifractal approach

Arun Ramanathan[1], Pierre-Antoine Versini[1], Daniel Schertzer[1], Remi Perrin[2], Lionel Sindt[2], and Ioulia Tchiguirinskaia[1]

[1]École des Ponts Paristech (ENPC), Laboratory of Hydrology Meteorology & Complexity
[2]SOPREMA

**Correspondence:** Arun Ramanathan (arun.ramanathan@enpc.fr)

**Abstract.** Hydrological applications such as storm-water management or flood design usually deal with and are driven by region-specific reference rainfall regulations or guidelines based on Intensity-Duration-Frequency (IDF) curves. IDF curves are usually obtained via frequency analysis of rainfall data using which the exceedance probability of rain intensity for different durations are determined. It is also rather common for reference rainfall to be expressed in terms of precipitation $P$, accumulated in a duration $D$ (related to rainfall intensity $\frac{P}{D}$), with a return period $T$ (inverse of exceedance probability). Meteorological modules of hydro-meteorological models used for the aforementioned applications therefore need to be capable of simulating such reference rainfall scenarios. This paper aims to address the three interrelated yet distinct research gaps: i) the general discrepancy between standard methods for defining reference precipitation and the strong multi-scale intermittency of precipitation, ii) lack of procedures to adapt multifractal precipitation modelling to specified partial statistical references, and iii) lack of proper multiscale tools to quantitatively estimate the effectiveness of such simulation procedures. To do accomplish these aims it does the following: i) proposing a procedure designed to tackle multi-scale intermittency head-on, based on extreme non-Gaussian statistics and scaling behaviour over two sub-ranges of time scales, due to the finite size of the earth, ii) defining a renormalization procedure for the multifractal model to make the simulations comply with the aforementioned partial statistical references, and finally iii) defining multiscale metrics to compare the simulated rainfall time series with those observed. The scope of this paper is that the baseline precipitation scenarios simulated by this procedure can be used as more realistic inputs into hydrological models for applications such as the optimal design of storm-water management infrastructure, especially green roofs. The multifractal cascade framework, since it incorporates physically realistic properties of rainfall processes (non-homogeneity or intermittency, scale invariance and extremal statistics) is utilized in the proposed procedure. Here we suggest a discrete-in-scale universal multifractal (UM) cascade based approach. Daily, Hourly and six-minute rainfall time series datasets (with lengths ranging from 100 to 15 years) over three regions (Paris, Nantes, and Aix-en-Provence) in France that are characterized by different climates are analyzed to identify scaling regimes and estimate corresponding UM parameters ($\alpha$,$C_1$) required by the UM cascade model. Suitable renormalization constants that correspond to the $P$,$D$,$T$ values of reference rainfall are used to simulate an ensemble of reference rainfall scenarios, and the simulations are compared with datasets. Although only purely temporal simulations are considered here, this approach could possibly be generalized to higher spatial dimensions as well.

**Keywords.** Multifractals, Non-linear geophysical systems, Cascade dynamics, Scaling, Hydrology, Stochastic rainfall simulations.

## 1 Introduction

Reference rainfall events characterized by amount of precipitation $P$, duration $D$ and return period $T$ are required for sizing storm-water management infrastructures such as conduits, retention basin, and even green roofs if considered as a storm-water management tool. For this purpose, designed hyetograms are traditionally used. They represent a huge simplification of the reference event: homogeneous and constant precipitation, triangle shape, etc. In reality, rainfall is quite commonly considered to be a stochastic variable due to the fact that rainfall process is complex and strongly dependent on initial conditions. Therefore reference rainfall events used for sizing should take into account this complexity. Nevertheless, availability of high-resolution observational datasets for rainfall especially over lengthy time periods and/or vast spatial areas is quite limited even today. Consequently, there have been several studies/attempts to stochastically produce rainfall time series and space-time fields as listed here: Simple point processes (Salas, 1993; Heneker et al., 2001), Cluster processes (Cowpertwait, 1994; Cameron et al., 2000b, a; Cowpertwait et al., 2011; Kaczmarska et al., 2014), Hybrid processes (Gyasi-Agyei and Willgoose, 1999; Onof et al., 2000; Li et al., 2012), and models that use the Monte Carlo method to generate hyetograms i.e. temporal distribution of rainfall intensity (Arnaud and Lavabre, 1999; Kottegoda et al., 2014). All these four model types are purely temporal. Markov chain (Wilks, 1998; Gao et al., 2020, 2021), and Non-parametric (Rajagopalan and Lall, 1999; Brandsma and Buishand, 1998; Mehrotra and Sharma, 2006; Kannan and Ghosh, 2013) models, on the other hand, simulate rainfall time series at a few distinct spatial points and can therefore be considered to be slightly more advanced than purely temporal models. Cell clusters (Wheater et al., 2000, 2005; Koutsoyiannis and Onof, 2001; Park et al., 2021), Modified turning band (Shah et al., 1996; Leblois and Creutin, 2013), Radar-based bead (Pegram and Clothier, 2001; Berenguer et al., 2011; Paschalis et al., 2013, 2014; Nerini et al., 2017) models can be considered are a bit more involved than the aforementioned models, however they do make some non-physical simplifying assumptions and are still not that parsimonious. Alternatively, there are other procedures utilizing point models (Cowpertwait et al., 1996; Gyasi-Agyei, 2005; Pui et al., 2012), and artificial neural networks (Burian et al., 2001; Gholami et al., 2015; Di Nunno et al., 2022) that generally deal with downscaling of rain fields from numerical weather prediction (NWP) models. Finally there are a few physically-based yet computationally simple and parsimonious models such as Non-homogeneous random cascades (Schertzer and Lovejoy, 1988, 1989, 2004a, b; Pathirana and Herath, 2002; Serinaldi, 2010; Gires et al., 2020) that are capable of taking into consideration the realistic spatio-temporal complexity of rainfall fields.

To make a literature-based assessment of these aforementioned modelling approaches in the context of using the resulting simulations as input for most hydrological applications (including the designing of rain-water management infrastructures) we consider eight characteristics of observed rainfall fields that if incorporated by the framework makes the simulations realistic: 1) Heterogeneity: Spatial Heterogeneity – rainfall is extremely variable with spatial location, especially at small spatial scales and Temporal Heterogeneity (intermittency) - rainfall time series at a single spatial location is extremely variable with time, especially at small time scales. 2) Physically based – the model represents the underlying process (at least abstractly) using

physically meaningful parameters (in a slightly more generalized framework, because it is stochastic rather than deterministic, with fractional rather than integer derivatives), 3) Nonlinearity – for instance, fields are not presumed to be additive, 4) Space-time complexity – both spatial and temporal variability/properties of the field can be considered simultaneously thereby incorporating possible space-time anisotropy, 5) Extreme statistics – extreme rainfall values are more frequent than usual resulting into strongly non-Gaussian statistics, 6) Statistical non-stationarity with the possibility of long-term memory – the statistical properties of the field being auto-correlated over larger temporal lags. The last two characteristics that are considered for the assessment make models practically attractive: 7) High Parameter parsimony – the model uses only a few parameters, 8) Low Computational complexity – the entire simulation procedure including parameter estimation is not too time consuming. The existence of simplifying physical principles such as universality help the frameworks in being highly parsimonious and computationally simple without compromising too much on the physical relevance of the simulations. Table. 1 shows a literature-based comparison of the desirable characteristics possessed by each model sub-classification. As shown in Fig. 1 most of the aforementioned models (10 out of 12) seem to be more focussed on computational and conceptual simplicity than on physics. Alternatives such as Universal Multifractal (UM) cascades that aren't computationally that complicated (compared to high-resolution Numerical Weather Prediction models that explicitly represent given atmospheric processes on a limited range of scale) therefore seem to be attractive choices especially since they are capable of representing fields with high spatio-temporal variability (Schertzer and Lovejoy, 1989; Ladoy et al., 1993; Tessier et al., 1996; Lovejoy and Schertzer, 2006, 2007; Schertzer et al., 2010; Schertzer and Lovejoy, 2011; Hoang et al., 2012; Lovejoy and Schertzer, 2013; Gires et al., 2013; Hoang et al., 2014). These UM cascade models needs only observational rainfall time series (not very data demanding) and are computationally simpler and parsimonious compared to the Radar-based bead method (Pegram and Clothier, 2001) mentioned earlier. Such UM-based procedures can also be directly extended to obtain space-time fields as well. Furthermore, the idea of space-time complexity in the UM framework is somewhat more generalized than it is in the Radar-based bead model (where spatial complexity and temporal complexity are dealt with separately rather than together).

The objective of this paper is to address three kinds of research gaps: i) a general discrepancy between standard procedures for defining reference precipitation and the strong multiscale intermittency of precipitation, ii) missing procedure to adapt multifractal precipitation modelling to given partial statistical references, and iii) missing procedure to assess the accuracy of the method. This is done by i) tackling multiscale intermittency head-on, based on extreme non-Gaussian statistics and scaling behaviour over two subranges of time scales, due to the finite size of the earth (which requires some adaptation of the multifractal modelling procedure), ii) defining a renormalizing procedure for the multifractal model to make the simulations fit with these partial statistical references, and iii) defining multiscale metrics to assess distance between (closeness of) two time series (observed and simulated) across time scales. This will enable the generation baseline precipitation scenarios that can be used as realistic inputs into hydrological models for applications such as the optimal design of storm-water management infrastructure, especially green roofs. Region-specific (single-site separately for three different sites/conurbations) reference rainfall time series (characterized by the required properties: P,D,T) that exhibit larger variability and intermittency over a wide range of time-scales (close to that of observed rainfall data) compared to traditional procedures (which often utilize uniform rainfall or synthetic hyetograms) that do not take into account the high temporal variability of rainfall fields (Qiu et al., 2021)

are therefore simulated here. It is worth noting that simulating just rainfall time series instead of space-time fields is justified because: i) the dichitomy (between time and space-time) is not as strong as usual for multifractal models because a multifractal time series can be seen as a temporal cut of a space-time multifractal field, ii) the aim of the present study is focused on storm-water management over a fixed (and rather small) spatial area such as a building roof (as mentioned earlier), and iii) the large-scale deployment of rainfall-runoff management technologies would instead require space-time models, obtained with the help of new and rather limited developments (as mentioned in i)). Section 2 discusses the different regions considered in France, their corresponding reference rainfall regulations and the observational datasets used. These rainfall datasets are analysed via multifractal techniques as shown in Section 3 to identify scaling regimes and corresponding UM parameters necessary to simulate rainfall. Section 4 gives a brief recollection about discrete-in-scale UM cascades, explains in detail the procedure used here to simulate reference rainfall scenarios, and finally defines four metrics to quantitatively compare the simulations with corresponding datasets. Finally, the conclusions of this study along with its limitations and some future scope (extension to higher dimensions and other regions) are discussed in Section 5.

## 2    Regions considered and observational datasets used

French regional storm-water management/discharge regulations are usually expressed in relation with some reference rainfall events expressed in terms of precipitation $P$, duration $D$, return period $T$ values. As shown in Table. 2 the $P,D,T$ values - for 3 different localities - display high variability, but this is not that surprising since these values correspond to reference rainfall and rainfall like many other geophysical fields exhibits high spatio-temporal variability. As seen from the $P,D,T$ combinations for Nantes and Aix-en-Provence it is very clear that these specifications are highly variable even within the same region considered and the corresponding hydrological designs have to take into account such high space-time variability of rainfall at least up to (and in fact more than) these legal constraints or regulations. Therefore, it is quite logical that the modelling technique to be used for stochastically simulating an ensemble of such highly variable reference rainfall scenarios should explicitly incorporate properties of heterogeneity and Non-Gaussian statistics among several other properties that the observed fields typically exhibit. The rainfall datasets/time-series (since the aim here is to simulate rainfall time series as explained in Section 1) used for the three regions (Paris, Nantes and Aix-en-Provence) were obtained from MeteoFrance (https://donneespubliques.meteofrance.fr/). Different time series were collected according to the time step (6-minute, hourly, daily). Figure. 2 shows the selected conurbations and their climatological rainfall data. These three regions were selected for this study as their monthly cumulative rainfall climatology (computed from daily data sets) are quite different from each other: while Paris receives around 40-60 mm monthly rainfall, Nantes receives a higher monthly rainfall from around 40-90 mm, Aix-en-Provence on the other hand receives a more variable monthly rainfall from around 10-80 mm. Cities are chosen here since storm-water management is more vital in urban areas due to their limited infiltration capacity. Information about the datasets used for each city/conurbation are given in Table 3. Since the proportion of data missing is low, replacing these values with zeros will probably not result in any significant change to the actual data. For the sake of simplicity, we shall henceforth

refer to the daily, hourly and 6-minutes datasets of Paris, Nantes and Aix as PD1, PD2, PD3, ND1, ND2, ND3, AD1, AD2, AD3 respectively.

## 3   Multifractal analysis of rainfall data

The concept of universality in complex systems states that only a few parameters (out of many) are relevant for defining the system since the same dynamical process is repeated scale after scale or the process interacts with many independent processes over a range of scales resulting in this reduction (Schertzer and Lovejoy, 1987; Schertzer et al., 1991; Schertzer and Lovejoy, 1997). In the UM framework only three parameters $\alpha$,$C_1$,$H$ (therefore referred to as UM parameters) are necessary. The three universal multifractal parameters have different geometrical and physical meanings. The degree of multifractality $\alpha$ defines the deviation from monofractality and its value is between 0 and 2 (if $\alpha = 0$ the process is mono-/uni- fractal with a unique fractal scaling exponent contrary to other cases ($\alpha \neq 0$), if $\alpha = 2$ the process has maximum multifractality with a larger spectrum of scaling exponents), codimension of the mean $C_1$ which describes the sparseness of the level of activity that dominantly contributes to the mean field ($C_1 = 0$ if the rainfall is homogeneous or in other words, if it always rains). The parameter $H$ (not exactly the same as Hurst's exponent but related to it) quantifies the deviation from a conservative process ($H = 0$), where the ensemble average of the field is conserved or in other words the ensemble average of the normalized field is 1. In a stochastic multifractal formalism, the $q$-th order statistical moment of rainfall $R_\lambda$ observed at a scale $l$ follows the multiscaling equation:

$$\langle R_\lambda{}^q \rangle = \lambda^{K(q)} \tag{1}$$

where $\lambda$ is the intermediate scale ratio or (temporal) resolution (ratio of the largest scale to the intermediate scale $l$), the equality sign is used here in a scaling sense, and the scaling exponent $K(q)$ is the scaling moment function that is scale-independent. For conservative UM, $K(q)$ depends only on the UM parameters as follows:

$$K(q) = \begin{cases} \frac{C_1}{\alpha - 1}(q^\alpha - q) & \forall \quad 0 \leq \alpha < 1, \quad 1 < \alpha \leq 2 \\ C_1 q \log q & \forall \quad \alpha = 1 \end{cases} \tag{2}$$

By computing the trace moments and double trace moments the function $K(q)$ and UM parameters can be empirically estimated (Schertzer and Lovejoy, 1987; Lavallee et al., 1993) as briefly discussed in the following two subsections. We consider each observational dataset to be a single sample (to avoid any reduction in the largest scale considered which may lead to different multifractal characteristics). However, there is a drawback due to this small sample size (i.e. $N_s = 1$, making the effective dimension equal to the dimension of the time series which is 1): the estimate of spectral slope $\beta$ is unreliable (coefficient of determination of the straight line fit is too low). Larger the sample size, better will be the estimate of spectral slope (better straight line fit). But increasing sample size with a fixed dataset length means that with more samples the length of each sample is smaller, implying that there is a reduction in the largest scale considered. This may in turn lead to a difference in multifractal characteristics. The TM analysis, on the other hand, does not have this disadvantage and the straight line fits are reasonably good and not too dependent on the number of samples. Therefore, TM analysis is simply more preferable/relevant compared

to spectral analysis or estimating how many samples would be ideal when using spectral analysis. Since $H = \frac{\beta + K(2) - 1}{2}$, consequently the $H$ values estimated using $\beta$ are also not very accurate. Therefore $H$ is estimated by considering the first order $(q = 1)$ un-normalized trace moment $\langle R_\lambda{}^1 \rangle$ (initially) assuming that the time series is non-conservative

$$\langle R_\lambda{}^1 \rangle = \lambda^{-H} \tag{3}$$

where once again equality sign is used for a possible asymptotic equivalence $(\lambda \to \infty)$.

It turns out that for all the datasets (from PD1 to AD3) the slope of a straight line fitted through a log-log plot of $\langle R_\lambda{}^1 \rangle$ vs. $\lambda$ is close to zero, implying $H \approx 0$ (as shown in Table. 4). Therefore, we proceed by assuming the observed rainfall time series used in this study are conservative.

### 3.1 Trace Moment (TM) Analysis

In the TM analysis (Tessier et al., 1993; Schertzer and Lovejoy, 1987, 1992) rainfall $R_\Lambda$ at the finest given (temporal) resolution or scale ratio $\left( \Lambda = \frac{largest\ scale}{smallest\ scale} \right)$ is averaged to obtain rainfall over coarser and coarser resolutions $R_\lambda$ ,where the intermediate scale ratio $\lambda$ is a decreasing integer power of $\lambda_1$ $(\lambda = \lambda_1{}^n, \Lambda = \lambda_1{}^N; n = N, \ldots, 0)$, which is the scale ratio of the elementary cascade step and usually equals 2:

$$R_{\lambda_1{}^n}(j) = \frac{1}{\lambda_1} \sum_{i=1}^{\lambda_1} R_{\lambda_1{}^{n+1}}(\lambda_1(j-1) + i); \quad j = 1, 2, \ldots, \lambda_1{}^n; \quad n = N - 1, \ldots, 0 \tag{4}$$

Since rainfall time series are multifractals their statistics follow the multiscaling equation Eq. (1), therefore the trace moments at coarser and coarser (temporal) resolutions $TM_\lambda = \frac{\langle R_\lambda{}^q \rangle}{\langle R_\lambda \rangle^q}$ when plotted vs. $\lambda$ in a log-log coordinate can be used to estimate the slope $K(q)$ of a fitted straight line. Figure. 3 shows the results of this analysis done on all the datasets (PD1 to AD3): there are two scaling regimes having distinct slope or $K(q)$ with a scaling break (the scale where $K(q)$ changes abruptly and distinctly) at around 2 to 4 weeks (the synoptic maximum). All these scaling ranges of both the first and second scaling regimes are tabulated in Table. 4. Henceforth the scaling moment functions of the first and second scaling regime are denoted as $K_1(q)$ and $K_2(q)$ respectively.

### 3.2 Double Trace Moment (DTM) Analysis

Although the TM analysis helps in estimating $K(q)$, it does not provide explicit estimates of UM parameters $\alpha, C_1$. To do this the DTM analysis (Lavallee et al., 1993) is used:

$$DTM_\lambda = \lambda^{\eta^\alpha K(q)} \tag{5}$$

where $\eta$ is the power to which the rainfall time series is raised. Eq. (5) suggests that when $K(q, \eta)$ vs. $\eta$ is plotted in log-log coordinates, the slope of a fitted straight line gives the estimate of $\alpha$, whereas $C_1$ is calculated using this $\alpha$ estimate and the y-intercept of the fitted straight line. While performing the usual DTM analysis it is found that the $\alpha$ estimates are larger than 2 (thereby exceeding the limits in Eq. 2) in the first scaling regime for all the datasets considered here. Generally this could

be due to two different issues: (i) an incorrect $\alpha$ estimation procedure, or (ii) an incorrect assumption about the processes conservativeness. However, for the datasets considered here the first possibility seems more likely due to the fact that the H estimates are negligibly small (as shown in Table. 4 and discussed earlier in section 3) and that Fourier analysis of these datasets are unreliable due to the small sample size chosen ($N_s = 1$). Therefore, to overcome this issue an iterative DTM procedure is used here. More technical details about this procedure is given in the Appendix A. Table. 4 shows the UM parameters estimated

using the 9 different datasets, while Fig. 4 shows the DTM based estimation procedure. The parameters for the first scaling regime and second scaling regimes are denoted by the subscripts 1 and 2 respectively. Although 3 different scaling breaks and 6 different pairs of $\alpha,C_1$ values are empirically estimated (3 pairs for each scaling regime) for each region, for simulating a reference rainfall scenario that corresponds to rainfall observed in the corresponding region only 1 scaling break and 2 pairs of $\alpha,C_1$ values (1 pair for each scaling regime) are necessary (since these values are not too dependent on the dataset used,

this choice is justified). The UM parameters estimated from the daily and six-minutes data are selected to be used for the first and second scaling regime in the simulations, whereas the median value of scaling breaks (out of the three scaling breaks estimated from daily, hourly and six-minutes datasets) are chosen. To confirm that this selection procedure does not result in any significant difference in the multifractal characteristics of the datasets and the corresponding simulations we compute the Multifractal Comparison Index (MCI) based on the difference in the theoretical maximum observable singularity from a

finite-sized sample $\gamma_s$ (Hubert et al., 1993; Douglas and Barros, 2003)

$$\text{MCI} = \frac{1}{6} \sum_{j=1}^{3} \sum_{i=1}^{2} \left| \gamma_{s,obs(j)}(i) - \gamma_{s,sel(j)}(i) \right| \tag{6}$$

based on the difference between UM parameter values observed from datasets and selected for simulations (as indicated by the subscripts $obs$ and $sel$) with the analytical expression

$$\gamma_s = \frac{C_1 \alpha}{\alpha - 1} \left( \left( \frac{1}{C_1} \right)^{\frac{\alpha - 1}{\alpha}} - \frac{1}{\alpha} \right) \tag{7}$$

with respect to $\alpha$ and $C_1$, the indices $i,j$ denote the scaling regime (first or second) and the dataset (6-minutes, hourly or daily) used respectively.

MCI is computed to be 0.03 for both Paris and Nantes, and 0.04 for Aix. These low values of MCI justify the aforementioned selection procedure. Although multifractal (statistical) analysis of observed rainfall in the three conurbations chosen by this study do not display any significant seasonality (as there is no scaling break around a few months time scale), there is a

210 clear evidence of a strong synoptic maximum (indicated by a scaling break around few weeks time scale) with corresponding changes in scaling behaviour as seen in Fig. 3. It is worth noting that this aforementioned absence of seasonality in multifractal characteristics could imply that the low frequency scaling regime's UM parameters are sufficient to represent seasonal variability (in cumulative precipitations - Fig. 2), whereas together with the high frequency scaling regime's UM parameters they are sufficient for reproducing well the statistics of different storm types (either convective or stratiform). This requires some

215 elaboration of the UM cascade process (as detailed in Section 4) to guarantee good agreement between observed and simulated rainfall over the full range of time scales.

## 4 Discrete-in-scale Universal Multifractal cascades

Multifractal cascade processes have strongly non-Gaussian statistics (e.g., fat-tailed distributions) and therefore are capable of generating structures of highly varying intensities. These cascades represent the atmospheric physical processes underlying rainfall generation in an abstract (Richardson's idea of energy transfer from large to small scales by random breakups of eddies) but explicit manner by the concept of scale-symmetry or scale-invariance (a property respected even by the Navier-Stokes equation used by state-of-the-art NWP models for operational weather forecasting). Therefore, these types of models can be considered as a bridge between purely statistical and purely physical models. Due to their multiplicative property the heterogeneity of the simulated field increase incrementally at smaller scales (making these models capable of generating scale-dependent rain rates as observed in nature). Although discrete-in-scale cascades consider scale-ratios that are integer powers of integers they exhibit better scaling properties and are pedagogically straightforward compared to continuous-in-scale cascades (Lovejoy and Schertzer, 2010a, b). Furthermore, for the current purpose of simulating rainfall time series anisotropic and vector generalizations are not very relevant. Therefore, the discrete-in-scale UM cascade model is used here to simulate an ensemble of rainfall scenarios for each region (and its corresponding $P,D,T$ specifications). The basic idea of discrete-in-scale cascades (Schertzer and Lovejoy, 1989, 2011) is to iteratively divide large-scale eddies (structures) using a constant integer scale (time-scale) ratio $\lambda_1$ (usually 2 as mentioned earlier) and multiplicatively distribute flux ($\varepsilon_\lambda$) to these sub-eddies randomly (stochastically). It is convenient to do this using an additive noise or generator $\Gamma_\lambda$ (generator) the exponential of which results in the multiplicatively modulated multifractal flux series at (temporal) resolution $\lambda$ (Schertzer and Lovejoy, 1989). To simulate universal multifractals (whose statistics are governed by Eqs. 1 and 2) this generator must satisfy

$$\langle \varepsilon_\lambda{}^q \rangle = \langle e^{q\Gamma_\lambda} \rangle = \lambda^{\frac{C_1}{\alpha-1}q^\alpha} \tag{8}$$

To do this an extremal Lévy random variable of index $\alpha$ and suitable amplitude (Pecknold et al., 1993; Gires et al., 2013) - that is a function of $C_1$ - is chosen as $\Gamma_\lambda$ (this generator generates the singularity $\gamma_\lambda$ corresponding to each sub-eddy). In the present context rainfall $R_\lambda$ accumulated in a given interval of time is the flux $\varepsilon_\lambda$. Such a simulated field when normalized by its ensemble average is canonically conserved (Schertzer and Lovejoy, 2004b).

### 4.1 Simulating reference rainfall scenarios

To have the same $P,D,T$ characteristics of the reference rainfall, a simulated rainfall series with largest (temporal) scale ($T_{sim}$) needs to have a specific number ($\rho$) of peak values of rainfall ($\geq P$) accumulated over specific durations ($D$) so that their return period $T = \frac{T_{sim}}{\rho}$. A simple way to do this is to multiply the simulated multifractal time series (with largest scale $T_{sim} = \rho T, \quad \forall \quad \rho \in Z^+$) by an appropriate renormalization constant (RC): $P$ divided by the $\rho$-th highest value in the multifractal series aggregated over duration $D$. Therefore, the simulated rainfall series are dependent on these $P,D,T$ values, resulting in 1 rainfall series for Paris, 11 rainfall series for Nantes, and 3 rainfall series for Aix. Since the observed datasets have two scaling regimes it is necessary to use a double cascade: a coarser (temporal) resolution cascade using parameters $\alpha_1, C_{1_1}$ for the first scaling regime and a finer (temporal) resolution cascade using parameters $\alpha_2, C_{1_2}$ for the second scaling regime. Let the

smallest scale observed and simulated be $\delta$ (here $\delta = 6$ minutes for all three regions). The largest temporal scale selected from
observed datasets $T_{sel}$ is related to the largest scale that can be simulated $T_{(s,sim)}$ (largest scale in simulated sample which is a power of $\lambda_1 = 2$ and $\leq T_{sim}$):

$$T_{s,sim} = \delta 2^{\lfloor \log_2 \frac{T_{sel}}{\delta} \rfloor}; \quad T_{s,sim} \leq T_{sim} \tag{9}$$

where $\lfloor x \rfloor$ denotes the integer part of $x$.

The coarse resolution cascade produces a multifractal time series $\varepsilon_{\lambda_B}$ where $\lambda_B = \left(\frac{T_{s,sim}}{T_{B,sim}}\right)$ and $T_{B,sim} = \delta 2^{\lfloor \log_2 \frac{T_{B,sel}}{\delta} \rfloor}$ is the simulated scaling break.Each rainfall value of this coarse time resolution multifractal series is now the parent structure of the second (fine time resolution) cascade that proceeds from $T_{B,sim}$ up to $\delta$. A multifractal time series $\varepsilon_{\lambda_\delta}$ where $\lambda_\delta = \frac{T_{s,sim}}{\delta}$ is thus finally produced by the double cascade simulation (DCS). The DCS is repeated a sufficient number of times (if $T_{s,sim} < T_{sim}$) to finally extract a time series $\varepsilon_{\Lambda_\delta}$ where $\Lambda_\delta = \frac{T_{sim}}{\delta}$ (here $\delta = 6$ minutes). The $\rho$-th highest value in a aggregated multifractal series $\bar{\varepsilon}$ ($\varepsilon_{\Lambda_\delta}$ aggregated to temporal resolution $\frac{T_s im}{D}$) when multiplied by RC should equal P. If we rank the values in series $\bar{\varepsilon}$ in decreasing order and call it $\bar{\varepsilon}_{DO}$, then the $\rho$-th value $\bar{\varepsilon}_{DO}(\rho)$ is the $\rho$-th highest value. Therefore, $RC$ is computed as

$$RC = \frac{P}{\bar{\varepsilon}_{DO}(\rho)} \tag{10}$$

The $RC$ computed using Eq. (10) when multiplied to $\varepsilon_{\Lambda_\delta}$ gives the final rainfall series that has characteristics corresponding to the reference rainfall. This entire procedure is repeated $n_e$ times to generate an $n_e$ member ensemble of possible reference rainfall scenarios (Fig. 5. schematically presents the whole simulation method). Here $n_e = 10$, i.e. an ensemble of 10 members (m1 to m10) are simulated.

Figure. 6 shows the reference rainfall simulations for Paris: both rainfall data and singularities can be compared from the figure. The maximum observed and simulated singularities are closer to each other than that of corresponding rainfall values. This may be attributed to the fact that singularities are less scale-dependent than rainfall and the observed and simulated rainfall have different scale ratios (resulting in the unreliability of comparison using parameters that are more scale dependent) since their largest scales are different in spite of their smallest scales being equal. Figure. 6 e) shows that the simulated rainfall (from one member of the ensemble: m10) obey the $P$,$D$,$T$ reference criterion for Paris region. To highlight the internal variability of the 10 reference rainfall scenarios simulated, events where exactly $P$ mm rainfall occur within $D$ hours duration are plotted separately in Figure. 6 f). Figures. A1 and A2 are the same as Fig. 6 f) but are for the different $P$,$D$,$T$ specifications of Nantes and Aix-en-Provence.

## 4.2 Comparing simulations with observational datasets

Four metrics possessing different properties have been defined to compare the stochastic simulations of rainfall to the actual datasets. The first metric is the Multifractal Comparison Metric (MCM), the second metric is the Rainfall Comparison Metric (RCM), the third metric is the Singularity Comparison Metric (SCM), whereas the fourth and final metric is the Codimension Comparison Metric (CCM). These metrics are defined with the general idea that lower metrics correspond to better simulations and vice-versa.

### 4.2.1 Multifractal Comparison Metric (MCM)

The MCM is a theoretical metric and is computed based on the maximum observable theoretical singularity $\gamma_s$ (Eq. 7) from a finite sample size $N_s \approx \lambda^{D_s}$ where $D_S$ is the sample dimension (Schertzer and Lovejoy, 1992) in each dataset and in each simulated member of the ensemble

$$\text{MCM} = \frac{1}{6} \sum_{j=1}^{3} \sum_{i=1}^{2} \left| \gamma_{s,obs}(i) - \frac{1}{10} \sum_{k=1}^{10} \gamma_{s,sim(k)}(i) \right| \tag{11}$$

where $j$ indicates the dataset used (daily, hourly or 6 minute), $i$ denotes the scaling regime (first or second), $k$ indicates the ensemble member. MCM is closely related to MCI defined earlier, the only difference between them is that MCM uses the UM parameters estimated from DTM analysis of simulated members, whereas MCI directly uses the UM parameters selected for simulations from the observed datasets. Therefore, as expected the MCM computed for all the simulations are very low (shown in Figure. 7) and close to MCI. Since both MCM and MCI depend only on the UM parameters (or in other words the multifractal characteristics of the series), they are scale-independent (this means that MCM of two time series of same or different temporal resolutions or lengths are not too different). Since renormalization does not affect the multifractal properties of a series, the MCM is independent of $P,D,T$. Lower values of MCM imply that the simulation has multifractal properties close to that of observed data.

### 4.2.2 Rainfall Comparison Metric (RCM)

RCM on the other hand is a more practical metric and is computed based on the highest rainfall value present in the dataset and in each simulation member:

$$\text{RCM} = \frac{1}{3} \sum_{j=1}^{3} \frac{\left| \max[R_{\Lambda(obs(j))}] - \frac{1}{10} \sum_{k=1}^{10} \max[R_{\Lambda(sim(k),j)}] \right|}{\max[R_{\Lambda(obs(j))}]} \tag{12}$$

where $\Lambda(obs(j)) = \frac{L_{obs(j)}}{\delta(j)}; \Lambda(sim(k),j) = \frac{L_{sim(k)}}{\delta(j)}; \quad \delta(j) = 1$ day,1 hour,6 minutes for $j = 1, 2, 3$, the indices $j, k$ have the same meaning as in MCM.

Lower values of RCM imply that the extreme behaviour of simulations are closer to that of the observed data. But RCM is sensitively dependent on scale and $P,D,T$. Therefore, as shown in Figure. 7 the RCM values are larger for cases where $\frac{P}{D}$ is larger. This might be due to the fact that the datasets used (since they are of shorter lengths) are not actually representative of these specific $P,D,T$ values that correspond to rainfall events that are more extreme (since probability of observing rarer events is higher in larger datasets).

### 4.2.3 Singularity Comparison Metric (SCM)

SCM is a metric that instead of comparing the actual time series compares the singularities corresponding to them, and is computed as:

$$\text{SCM} = \frac{1}{3}\sum_{j=1}^{3} \frac{\left| \max[\gamma_{\Lambda(obs(j))}] - \frac{1}{10}\sum_{k=1}^{10}\max[\gamma_{\Lambda(sim(k),j)}] \right|}{\max[\gamma_{\Lambda(obs(j))}]} \tag{13}$$

where $\gamma_{\Lambda(obs(j))} = \frac{\log R_{\Lambda(obs(j))}}{\log \Lambda(obs(j))}$; $\gamma_{\Lambda(sim(k),j)} = \frac{\log R_{\Lambda(sim(k),j)}}{\log \Lambda(sim(k),j)}$, the indices $j,k$ have the same meaning as in MCM.

Lower values of SCM imply that the simulations are closer to the observations (after reducing the effect of scale-dependence on the comparison) since the singularities corresponding to the simulations and the singularities corresponding to the observations are close to each other. Although SCM is scale-dependent it is less sensitive to scale than RCM; moreover SCM is also dependent on $P,D,T$. Therefore, SCM values of all simulations are low ($\leq 0.15$) even for cases where $\frac{P}{D}$ is larger as shown in Figure. 7.

### 4.2.4 Codimension Comparison Metric (CCM)

The main drawback of the MCM, RCM and SCM are that they focus only on either the maximum rainfall values or the maximum singularities. On the contrary, a range of values rather than threshold values can be used. For instance, the codimension of singularity $c(\gamma)$ takes into account a range of singularities larger than $\gamma$. Following Schertzer and Lovejoy 1987:

$$\Pr[R_\lambda \geq \lambda^\gamma] \approx \lambda^{-c(\gamma)} \tag{14}$$

meaning that $c(\gamma)$ can be obtained as the negative of the slope of a straight line fitted to log-log plot of $\Pr[R_\lambda \geq \lambda^\gamma]$ with respect to $\lambda$. Equation. 14 (where $\approx$ indicates an asymptotic equivalence) implies that $c(\gamma)$ is almost scale independent and any metric defined using it should also be not very scale-sensitive. The CCM is defined as

$$\text{CCM} = \frac{1}{3n}\sum_{j=1}^{3}\sum_{i=1}^{n}\left| c_{obs(j)}\big(\gamma_{\Lambda(obs(j))}(i)\big) - \frac{1}{10}\sum_{k=1}^{10} c_{sim(j,k)}\big(\gamma_{\Lambda(obs(j))}(i)\big) \right| \tag{15}$$

where $\gamma_{\Lambda(obs(j))}(i) = \min[\gamma_{\Lambda(obs(j))}] + \frac{1}{n}(i-1)(\max[\gamma_{\Lambda(obs(j))}] - \min[\gamma_{\Lambda(obs(j))}])$, the indices $j,k$ have the same meaning as in MCM, whereas $i$ indexes the singularities (here $n=10$ singularities are used for the comparison procedure).

The CCM is dependent on $P,D,T$ via the singularities $\gamma_\Lambda$, therefore in an almost scale-independent manner. As shown in Fig. 7, the SCM and CCM values are consistently low implying that it is possible to simulate reference rainfall ensembles characterized by the required properties $(P,D,T)$ while taking into account temporal variability. It is worth noting here that autocorrelation or its inverse Fourier transform i.e. spectral density are generally just second order statistics. Comparing the scaling moment function $K(q)$ for $q=2$ of observed and simulated rainfall is the same as comparing their respective spectra and therefore their autocorrelation. The CCM compares $c(\gamma)$ instead of $K(q)$ since they are just the Legendre transforms of each other and each order of singularity $\gamma$ corresponds to an order of statistical moment $q$. Therefore, the CCM is a more generalized metric as it readily considers the second order statistics and more.

## 5 Discussion

The $\alpha, C_1$ estimates for the second scaling regime and the scaling breaks listed in Table. 4 are quite comparable with those of earlier studies (Hubert et al., 1993; Ladoy et al., 1993). These breaks in temporal scaling can be attributed to the synoptic maximum (Tessier et al., 1996) or in other words the lifetime of planetary scale atmospheric structures. The similarity of scaling breaks observed in all the datasets justify the dependence of scaling break on the value of the largest planetary spatial scale (and its corresponding eddy turnover time or lifetime). Further more like in earlier studies (Hoang et al., 2014) the negligible $H$ estimates suggest that the process is conservative (in both scaling regimes). It can be seen that while the first scaling regime (low frequency) has a larger $\alpha$ and smaller $C_1$, the reverse is true for the second (high frequency) scaling regime. A similar pattern seems to be followed for all the three conurbations (irrespective of the dataset used) considered in this study. For a conservative process, this seemingly inverse relation between $\alpha$ and $C_1$ could be reasoned as follows: a larger $C_1$ value implies that the processes contributing dominantly to the mean occur rarely, since the probability of occurrence of singularities contributing to the mean is the highest this in turn implies that other singularities occur even more rarely or in other words the range of singularities is rather limited resulting in the process having a reduced degree of multifractality (i.e. smaller $\alpha$ values). On the other hand the smaller $C_1$ values (close to 0) in the low frequency (large time scale) scaling regime could be because at larger time scales ($>$ synoptic maximum) it can be expected to rain almost always. Comparing the UM parameter estimates (in the corresponding scaling regimes) of all the 3 conurbations, it can be seen that they are somewhat similar to one another. Based on the above discussion it seems that the rainfall can be considered to be the most intermittent (temporally heterogeneous) i) over smaller time scales in Aix, closely followed by Paris and finally by Nantes, and ii) over larger time scales in Aix, closely followed by both Paris and Nantes. This might explain (at least partially) why the reference rainfall rules for Aix seem to be too focused on extreme rainfall events (as seen in Table. 2).

In addition to the assessment of multi-scale properties of the simulated time series (the comparison metrics defined and discussed in the previous section), a few words can be said about the ability of the developed method to reproduce rainfall seasonality. A simple way to compare the seasonality of simulated and observed rainfall is to compute the respective monthly cumulative climatology (as done in Fig.2b) and estimate the number of months $n_{ms}$ between the occurrence of maximum and minimum monthly precipitation. From Fig.2b it can be seen that $n_{ms,obs}$ for all the three conurbations is 2, and from the simulated scenarios it is found that $n_{ms,sim}$ for Paris, Nantes and Aix are $2.3, 1.8, 2.6$ respectively. A simple metric $\frac{|n_{ms,obs} - n_{ms,sim}|}{12}$ can then be computed (which when close to 0 implies that seasonality in simulated rainfall is close to that observed) which for Paris, Nantes and Aix is computed to be $0.025, 0.017, 0.05$, thereby confirming that the simulations have reasonably realistic seasonality features.

## 6 Conclusions

Even though several earlier studies have attempted to simulate rainfall using a UM approach, we are unaware of UM-based studies that have proposed procedures to simulate reference rainfall scenarios. A novel method is proposed here to simulate reference rainfall scenarios that are indispensable for hydrological applications such as designing green roofs and other generic

storm-water management devices. The suggested discrete-in-scale Universal Multifractal cascade based method is used here to stochastically simulate an ensemble of reference rainfall scenarios (with rainfall events exceeding or equal to $P$ mm within $D$ hours duration having a return period of $T$ years) as specified by regional storm-water management regulations for three conurbations in France. The extreme variability of $P,D,T$ values which is a direct result of the extreme space-time variability of precipitation and underlying atmospheric processes, not only justifies but also makes the choice of UM framework rather crucial in producing computationally cheap, physically realistic reference rainfall ensembles. Furthermore, four new metrics are proposed to quantify the performance of the suggested procedure and analyse their effectiveness. Three (MCM, SCM and CCM) out of the four metrics (which are not too scale-dependent) seem to indicate that the simulations are good. CCM being almost scale-independent, and utilizing a range of values rather than just maxima for comparison seems to be the most reliable comparison metric. Therefore, the consistently low CCMs show that the proposed method is indeed an attractive choice to stochastically simulate physically-based reference rainfall scenarios. Although only purely temporal, discrete-in-scale, conservative simulations over three locations (Paris, Nantes, Aix) are considered in this study, this approach could possibly be generalized to spatio-temporal, continuous-in-scale, non-conservative simulations over other locations as well. While it is true that the proposed approach is for hydrological applications such as designing green roofs for rain-water management, observational data of not only rainfall but also discharge from the green roof will be necessary to validate the entire hydro-meteorological modelling approach. This would require the setting up experimental green roof prototypes designed using green roof models capable of simulating hydrological behaviour of both substrate and drainage layers with reference rainfall scenarios as input, and defining metrics that quantify compliance to regulations. These prototypes can then be monitored to estimate how much they comply with discharge rules via the compliance metrics. All these elements will therefore be subjects of separate publications in future. Finally, it is worth noting that the suitability of the UM framework in simulating reference rainfall scenarios and in several other geophysical applications (Ramanathan et al., 2018, 2019; Ramanathan and Satyanarayana, 2019, 2021; Ramanathan et al., 2021), illustrates that this generalized, physically based, computationally simple framework could probably be ideal for framing reference rainfall regulations guiding hydrological applications/designs.

**Appendix A**

To get more accurate $\alpha$ estimates for the first scaling regime an iterative DTM procedure is implemented here. Following earlier studies (Hoang et al., 2012) the idea of this procedure is to estimate $\eta_{\min} = (\frac{C_{\Sigma}}{C_1})^{\frac{1}{\alpha}} \max[1, \frac{1}{q}]$ and $\eta_{\max} = (\frac{1}{C_1})^{\frac{1}{\alpha}} \min[1, \frac{1}{q}]$: first using an initial guess of $\alpha, C_1$ (based on initial guesses of $\eta_{\min}$ and $\eta_{\max}$) then subsequent $\eta$ range and $\alpha, C_1$ estimates are obtained in each iteration until there is no longer any change in the $\eta$ range and therefore the $\alpha, C_1$ estimates. The codimension (difference between the dimension of the embedding space and that of the dimension of the set under consideration) of non-zero rainfall support $C_{\Sigma} = 1 - D_{\Sigma}$ (here $D_{\Sigma}$ is the fractal dimension of rainfall greater than the minimum threshold considered). However, the procedure used here is slightly modified: instead of searching for both $\eta_{\min}$ and $\eta_{\max}$ simultaneously in each iteration, the current procedure fixes $\eta_{\min}$ as a constant value (here it is initially 1) and obtains different $\eta_{\max}, \alpha, C_1$ values in each iteration. If the $\alpha$ estimate is still $> 2$ or if the $\alpha$ values keep changing even after a certain number of iterations, $\eta_{\min}$ is

slightly reduced and the whole procedure is repeated. A $q$ value of $0.8$ is used here so that the usable range of $\eta$ is larger (since the multifractal phase transition due to divergence of moments is more delayed) resulting in more reliable $\alpha, C_1$ estimates.

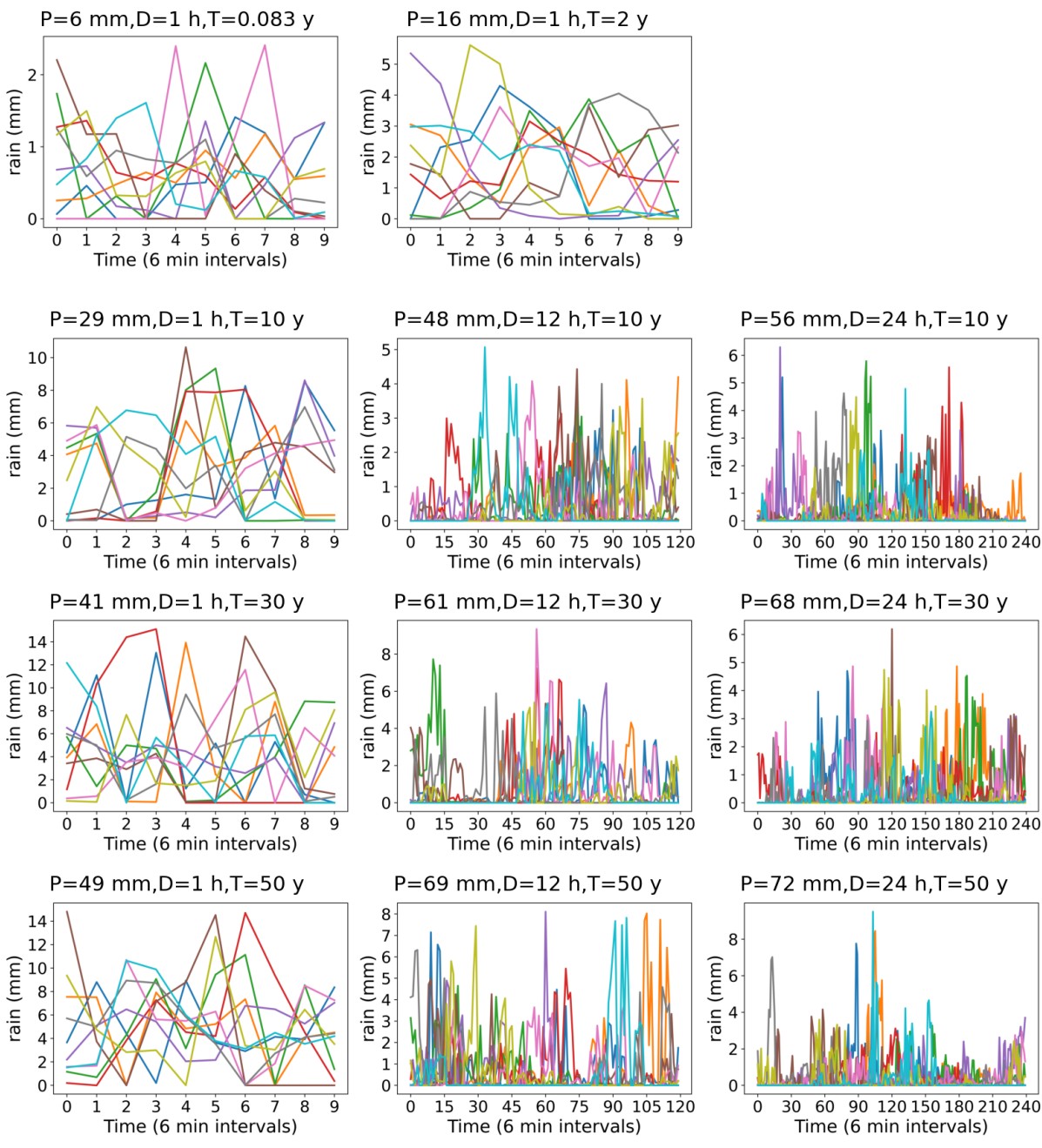

**Figure A1.** Ten rainfall scenarios (indicated by different colours) for Nantes with $P$ mm rainfall in $D$ hours (events such as these or more severe than these occur with a return period of $T$ years).

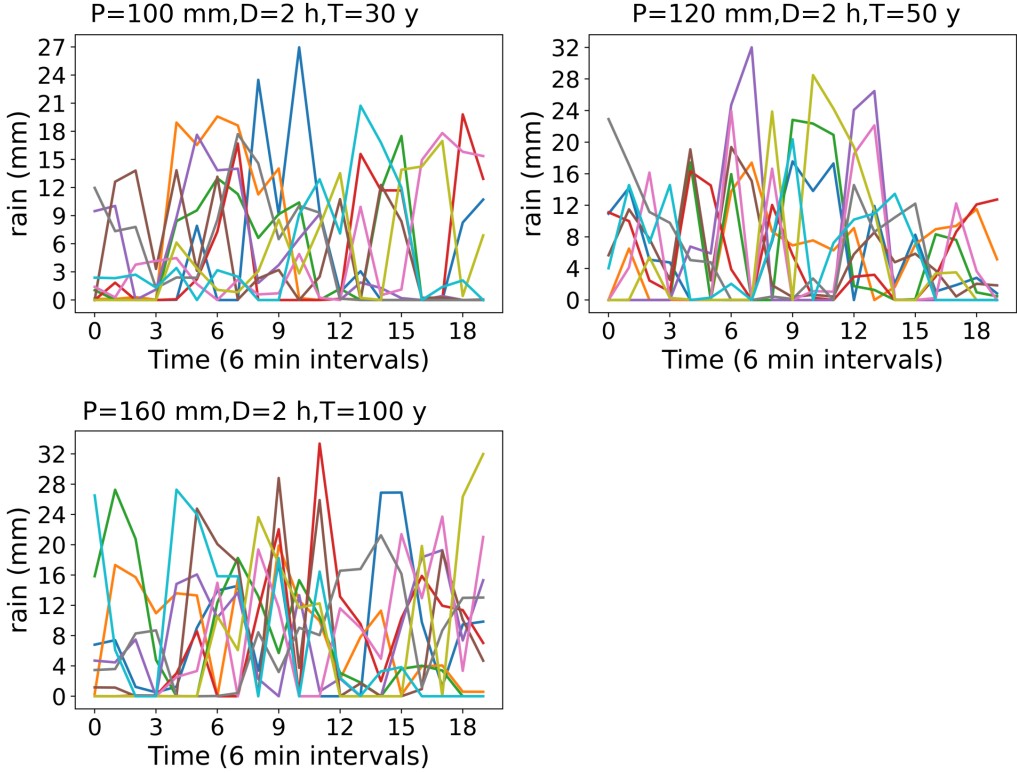

**Figure A2.** Ten rainfall scenarios (indicated by different colours) for Aix-en-Provence with $P$ mm rainfall in $D$ hours (events such as these or more severe than these occur with a return period of $T$ years).

*Author contributions.* A.R. performed the study and prepared the manuscript, P.A.V. and D.S. supervised the work, I.T. provided vital suggestions, R.P. and L.S. helped in gaining insights into commercial hydro-meteorological applications

*Competing interests.* The authors declare that no competing interests are present

*Acknowledgements.* The first author acknowledges SOPREMA for funding his post-doctoral research at HM&Co, ENPC. The authors acknowledge MeteoFrance for providing insitu rainfall observational datasets used in this study.

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

**Table 1.** Comparison of different stochastic rainfall modelling procedures based on literature.

| Models | Desirable Features | # of Parameters | Selected References |
|---|---|---|---|
| Simple point process | Computational simplicity | 5 | Salas (1993); Heneker et al. (2001) |
| Cluster processes | Computational simplicity | ≥5 | Cowpertwait (1994); Cameron et al. (2000a, b); Cowpertwait et al. (2011); Kaczmarska et al. (2014) |
| Hybrid processes | Computational simplicity | >5 | Gyasi-Agyei and Willgoose (1999); Onof et al. (2000); Li et al. (2012) |
| Monte Carlo based | Heterogeneity, Extreme statistics, Computational simplicity | ≥ 3 | Arnaud and Lavabre (1999); Kottegoda et al. (2014) |
| Markov chain | Non-stationarity, Heterogeneity, Computational simplicity | 4 | Wilks (1998); Gao et al. (2020, 2021) |
| Non-parametric | Extremal statistics, Nonstationary, Heterogeneity | 0 | Rajagopalan and Lall (1999);Brandsma and Buishand (1998); Mehrotra and Sharma (2006); Kannan and Ghosh (2013) |
| Point models | Computational simplicity | ≥ 5 | Cowpertwait et al. (1996); Gyasi-Agyei (2005); Pui et al. (2012) |
| Artificial neural netwroks | - | Varies | Burian et al. (2001); Gholami et al. (2015); Di Nunno et al. (2022) |
| Cell cluster | Computational simplicity | 7 | Wheater et al. (2000, 2005) Koutsoyiannis and Onof (2001); Park et al. (2021) |
| Modified Turning Band | Computational simplicity | 8 | Mellor (1996); Leblois and Creutin (2013) |
| Radar-based bead | Heterogeneity, Scale symmetry, Extremal statistics, Nonstationary, Computational simplicity | 4 | Pegram and Clothier (2001); Berenguer et al. (2011); Paschalis et al. (2013, 2014); Nerini et al. (2017) |
| Nonhomogeneous random cascade | Heterogeneity, Scale symmetry, Nonlinearity, Space-time complexity, Extremal statistics, Nonstationary, Computational simplicity | 2 | Schertzer and Lovejoy (1988, 1989, 2004a, b) Pathirana and Herath (2002); Serinaldi (2010); Gires et al. (2020) |

**Table 2.** Variability of reference rainfall regulations in the three regions considered by this study.

| Region | Duration $D$ (hours) | Return period $T$ (years) | Precipitation $P$ (mm) |
|---|---|---|---|
| Paris | 4 | 0.5 | 16 |
| Nantes | 1 | $\frac{1}{12}$ | 6 |
| | 1 | 2 | 16 |
| | 1,12,24 | 10 | 29,48,56 |
| | 1,12,24 | 30 | 41,61,68 |
| | 1,12,24 | 50 | 49,69,75 |
| Aix-en-Provence | 2 | 30 | 100 |
| | 2 | 50 | 120 |
| | 2 | 100 | 160 |

**Table 3.** Temporal Resolution, Length and Percentage of Missing data of rainfall datasets used in this study.

| Region | Dataset (time resolution) | Length $L_{obs}$ (years) | % Missing |
|---|---|---|---|
| Paris | PD1 (daily) | 100 (1921 - 2020) | 0 |
| | PD2 (hourly) | 28 (1993 - 2020) | 0.3 |
| | PD3 (6 minutes) | 15 (2006 - 2020) | 0.6 |
| Nantes | ND1 (daily) | 75 (1946 - 2020) | 0 |
| | ND2 (hourly) | 28 (1986,1994 - 2020) | 0.7 |
| | ND3 (6 minutes) | 15 (2006 - 2020) | 0.1 |
| Aix-en-Provence | AD1 (daily) | 60 (1961 - 2020) | 0 |
| | AD2 (hourly) | 28 (1993 - 2020) | 0.7 |
| | AD3 (6 minutes) | 15 (2006 - 2020) | 0.17 |

**Table 4.** UM parameter estimates for first and second scaling regimes from different datasets (PD1 to AD3), the scaling regimes and parameters selected for simulating rainfall over each corresponding region. $H$ values are not included in the selected parameters and assumed to be zero since these rain time series seem to be almost conservative (both $H_1$ and $H_2$ are close to zero).

| Region | Dataset | Scaling Regimes | $\alpha_1,$ $\alpha_2$ | $C_{1_1},$ $C_{1_2}$ | $H_1,$ $H_2$ | Selected for simulations Scaling Regimes | $\alpha_1,$ $\alpha_2$ | $C_{1_1},$ $C_{1_2}$ |
|---|---|---|---|---|---|---|---|---|
| Paris | PD3 | 15 years - 17 days | 1.97 | 0.03 | -0.00002 | 100 years - 21 days | 1.89 | 0.02 |
| | | 17 days - 6 mins | 0.56 | 0.45 | 0.002 | 21 days - 6 mins | 0.56 | 0.45 |
| | PD2 | 28 years - 21 days | 1.84 | 0.03 | 0.0002 | | | |
| | | 21 days - 1 hour | 0.55 | 0.48 | -0.003 | | | |
| | PD1 | 100 years - 32 days | 1.89 | 0.02 | 0.00008 | | | |
| | | 32 days - 1 day | 0.71 | 0.37 | -0.0007 | | | |
| Nantes | ND3 | 15 years - 17 days | 1.85 | 0.03 | 0.002 | 75 years - 21 days | 1.7 | 0.02 |
| | | 17 days - 6 mins | 0.69 | 0.38 | 0.002 | 21 days - 6 mins | 0.69 | 0.38 |
| | ND2 | 28 years - 21 days | 1.86 | 0.02 | 0.0002 | | | |
| | | 21 days - 1 hour | 0.59 | 0.42 | -0.0007 | | | |
| | ND1 | 75 years - 32 days | 1.7 | 0.02 | 0.00008 | | | |
| | | 32 days - 1 day | 0.65 | 0.35 | 0.002 | | | |
| Aix-en-Provence | AD3 | 15 years - 34 days | 1.79 | 0.04 | 0.00008 | 60 years - 32 days | 1.8 | 0.03 |
| | | 34 days - 6 mins | 0.51 | 0.48 | 0.0035 | 32 days - 6 mins | 0.51 | 0.48 |
| | AD2 | 28 years - 21 days | 1.76 | 0.06 | -0.00007 | | | |
| | | 21 days - 1 hour | 0.48 | 0.55 | 0.005 | | | |
| | AD1 | 60 years - 32 days | 1.8 | 0.03 | -0.0003 | | | |
| | | 32 days - 1 day | 0.49 | 0.54 | 0.0002 | | | |

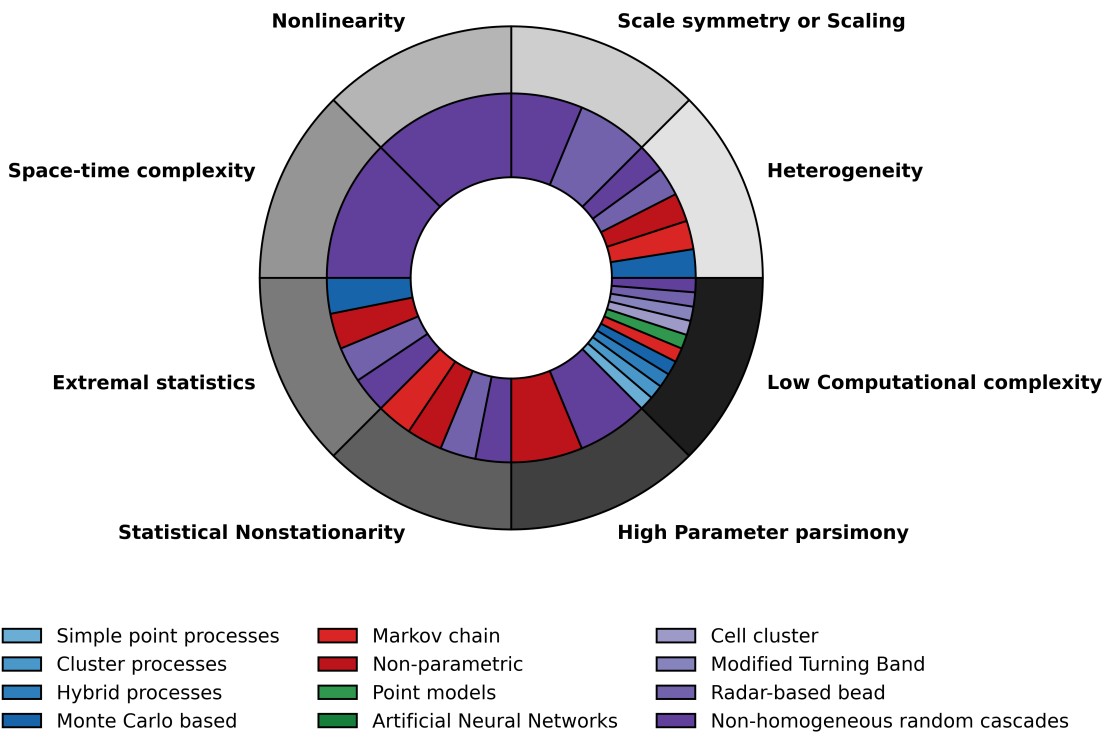

**Figure 1.** Outer-ring: Desirable characteristics in stochastic high-resolution rainfall simulation models. Inner-ring: Models that possess these characteristics (based on Table 1). Models with ≤3 parameters are considered here to possess High Parameter parsimony. Non-homogeneous random cascade models seem to possess all the desirable properties.

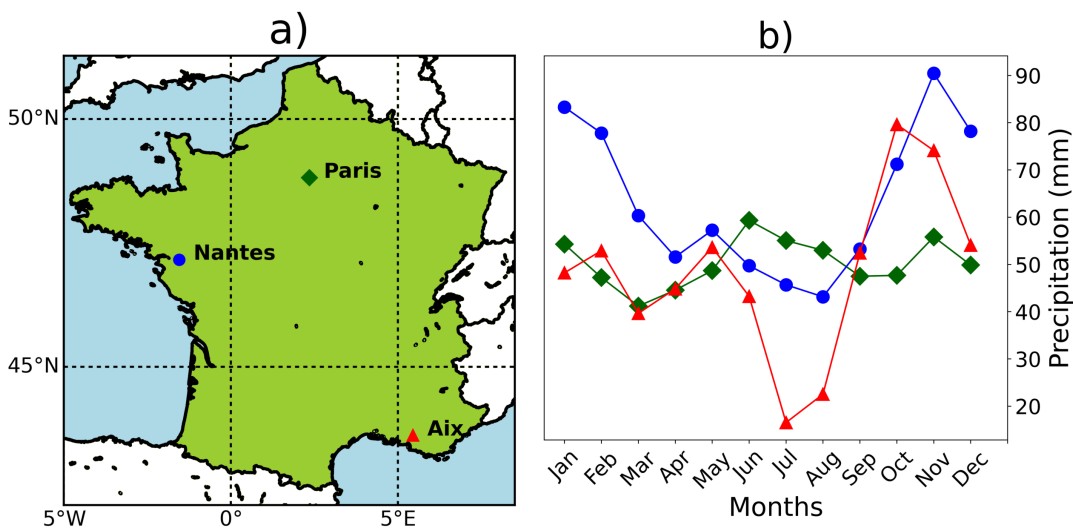

**Figure 2.** a) The three chosen cities/conurbations in mainland France, and b) their monthly cumulative precipitation climatology (using PD1, ND1 and AD1 datasets).

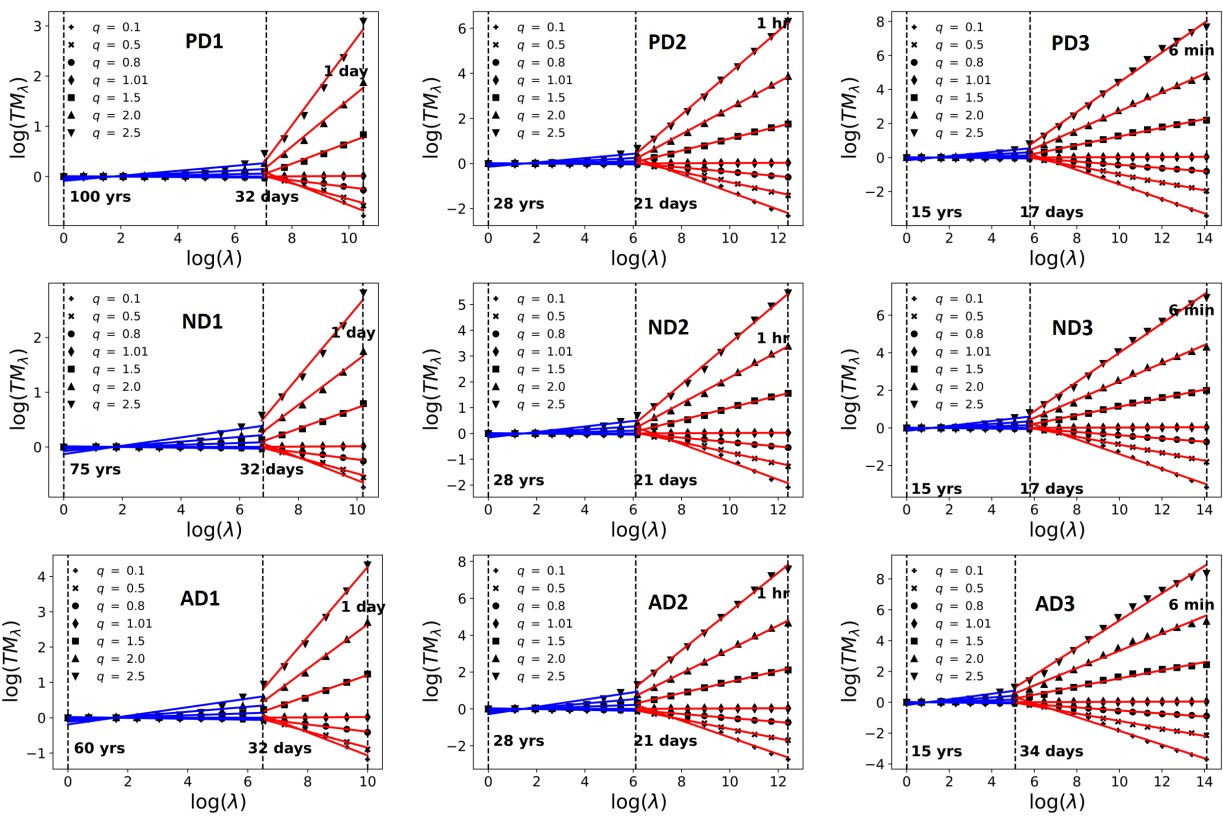

**Figure 3.** Trace Moment Analysis of accumulated rainfall data. Top Row: Paris: PD1, PD2, PD3; Middle Row: Nantes: ND1, ND2, ND3, and Bottom Row: Aix: AD1, AD2, AD3. The first scaling regime is shown in blue whereas the second scaling regime is shown in red.

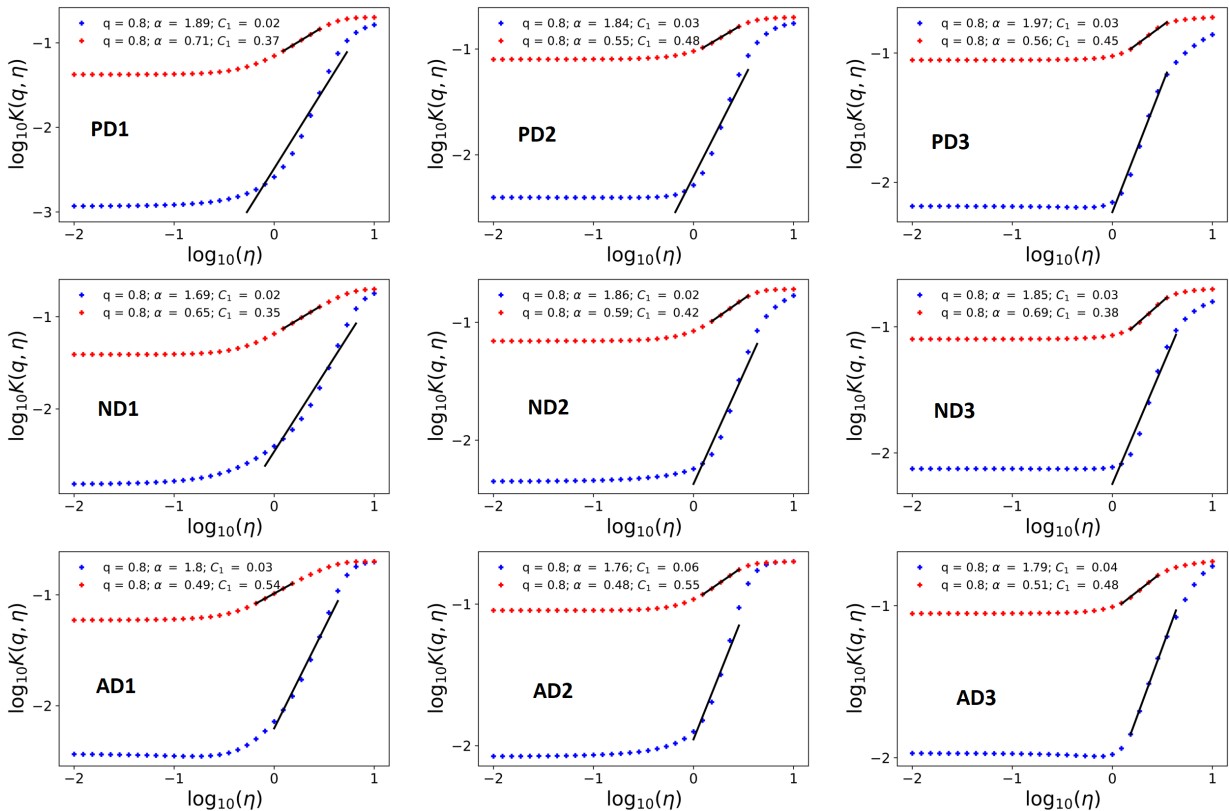

**Figure 4.** Double Trace Moment Analysis of accumulated rainfall data to obtain UM parameter estimates. Top Row: Paris: PD1, PD2, PD3; Middle Row: Nantes: ND1, ND2, ND3, and Bottom Row: Aix: AD1, AD2, AD3. The first scaling regime is shown in blue whereas the second scaling regime is shown in red.

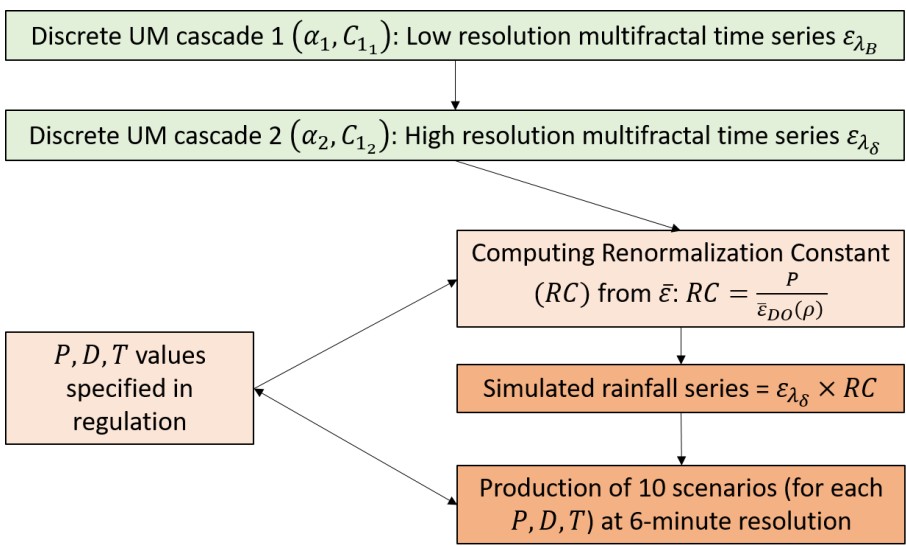

**Figure 5.** Schematic illustration of the simulation procedure used in this study to generate reference rainfall scenarios.

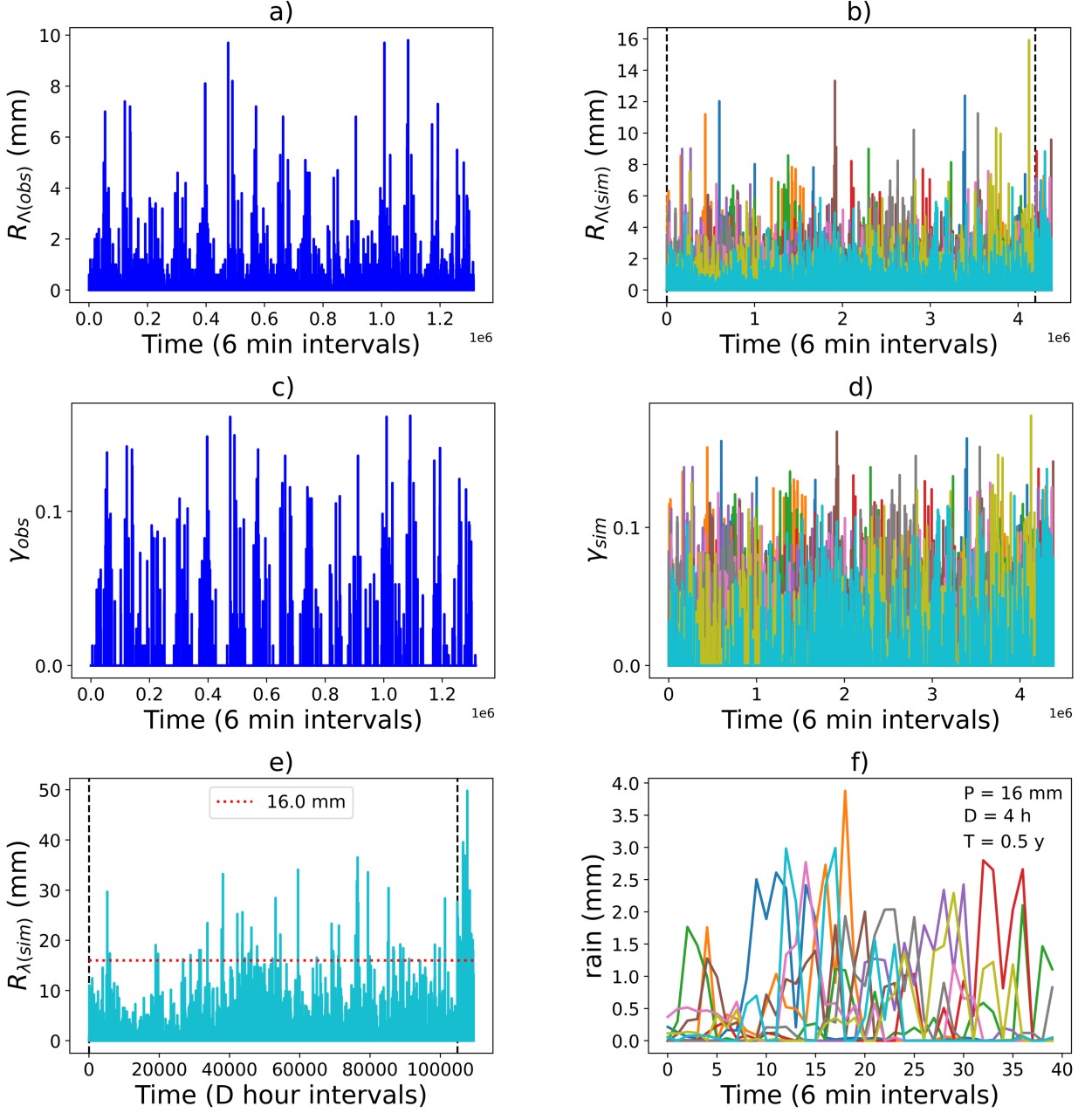

**Figure 6.** Paris reference rainfall scenarios ($P = 16$ mm, $D = 4$ hours, $T = 0.5$ years). a) Rainfall and c) corresponding singularities from observational dataset PD3; b) Rainfall, d) corresponding singularities, e) aggregated rainfall from member m10 and f) events with 16 mm cumulative rainfall in 4 hours duration from the ensemble double cascade simulation.

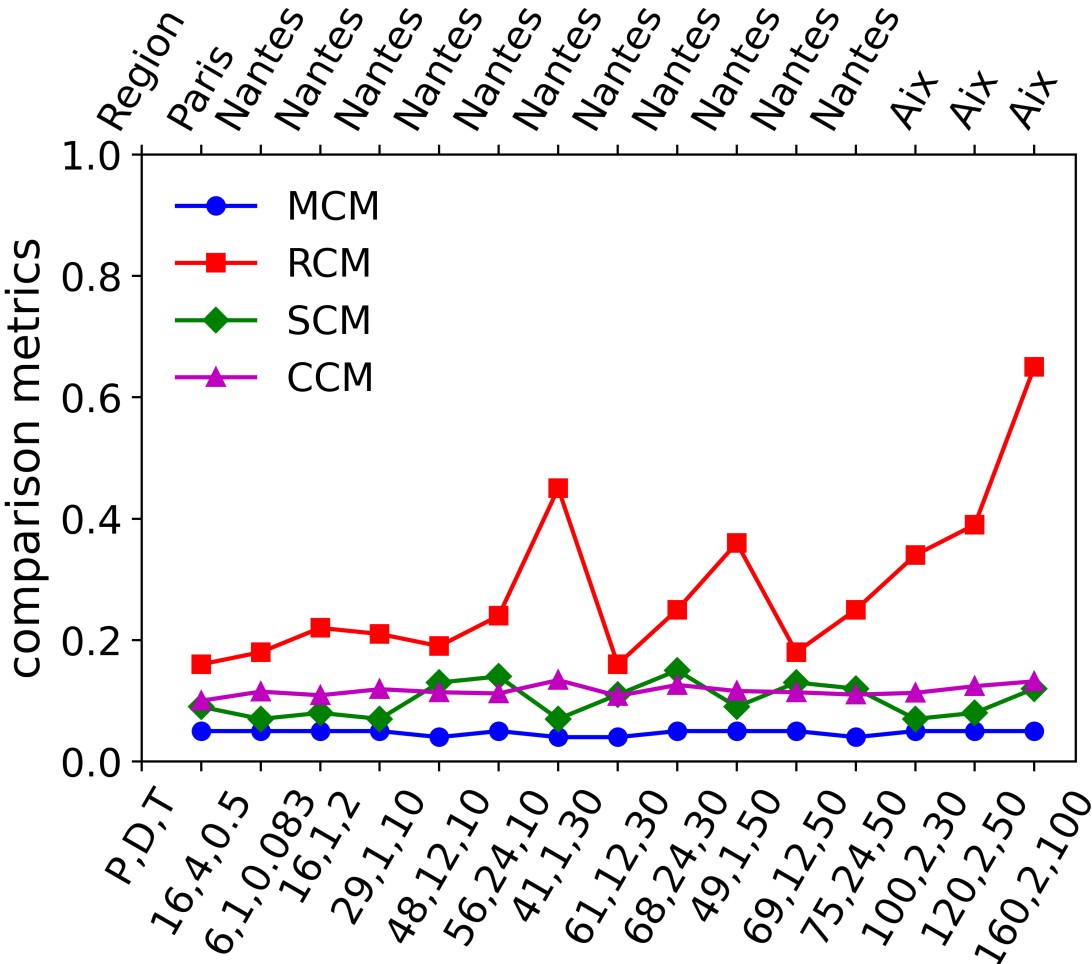

**Figure 7.** Multifractal Comparison Metric, Rainfall Comparison Metric, Singularity Comparison Metric and Codimension Comparison Metric for all the different reference rainfall simulations. $P$,$D$,$T$ are in units of mm, hours and years respectively.