# Peer review of "Stochastic simulation of reference rainfall scenarios for hydrological applications using a universal multifractal approach"

_Hydrology and Earth System Sciences, 2021_

## Referee Comment (RC1)

L7: research gap is not clear.

L21-23: it is not clear to me why 'uniform rainfall or … for such purposes' directly leads to the conclusion 'stochastic simulation of reference rainfall events is necessary'. At least one sentence (to discuss the lack of measurement data …) is missing.

L23: Does it refer to several types of single-site stochastic rainfall models or space-time models? Event-based simulation or continuous simulation? Please clarify.

L25: This paragraph is describing 'several types of stochastic rainfall models', but I am confused about why a model related to river discharge is described in detail. I get the point that one module in this 'Hydrogram model' is to generate hyetograms. If it is the case, I am wondering whether it is better to directly state that one of those types is the Monte Carlo method instead of the 'Hydrogram model'.

A general comment on this whole paragraph (L38) & Table 1:

The idea of creating this table looks good. However, more references would be needed. The latest one mentioned in this 'literature-based assessment' is in 2004, which is almost 20 years ago. In references for several model types for example in hybrid processes-based models, all references mentioned were published 20 years ago. Although those reference papers mentioned by the authors are very useful, it is necessary to update the reference list.

L38: does it refer to spatial rainfall field or spatially averaged rainfall time series?

L38: Does 'modelling approaches' refers to those modelling approaches for getting a realistic rainfall field for most hydrological applications or specifically for sizing storm-water management infrastructures?

L40: All these characteristics are '… makes the simulations physically relevant/realistic' while 'physically based' is specifically mentioned again in the second criteria 'Physically-based – the simulations rely on physical principles. What's the difference between these two? At least, the meaning of 'simulations rely on physical principles' in the second criteria need to be clarified.

L56: The research gap is not ever mentioned until now. It would be much better to clearly state it in the manuscript.

L57: Single-site or multi-sites or spatial-temporal rainfall field or something else? please clarify.

L59-61: Ok. this seems to be the research aim, which gives me the impression that the authors are implying that: Using any existing stochastic rainfall models, it is NOT possible to "simulate reference rainfall ensembles characterized by P, D, T while exhibiting temporal variability and intermittency close to that of observed rainfall data".

If my impression is correct, the authors were taking this as a research gap. However, this gap was not discussed or mentioned in the introduction section.

I suggest revising the whole section and adding more essential information to it to make this introduction section easier to follow.

L78: resolution (space and time)?

L120: are TM analysis and DTM analysis newly developed in this study? This is not clear to me.

L170: Is scale-symmetry able to be represented by other model types? From Table 1 in this manuscript, at least the Radar-based method can represent this, which can directly give us a space-time field. Why choose the multifractal theory instead of a radar-based method (see Pegram's paper as mentioned by the authors)? It will be helpful to justify the choice and include this discussion (probably in the introduction section).

L172: The authors mention that these types of models can be considered as a bridge between purely statistical and purely physical models. I am wondering whether these features have been in this cascade model used in this study: seasonality (or different types of storm event).

L176: It seems that the study presented here is to simulate temporal rainfall as the author mentioned in the abstract, but 'rainfall fields' convey a message that a time-varying spatial field is generated. Not entirely sure if this word is widely used when time series is generated. If not, please change this word.

L180: Does it always refer to temporal resolution?

L225: Ok, I can see that these metrics might be useful. Besides these, I am wondering whether the autocorrelation of simulated events is well reproduced. This feature plays a critical role in dominating a catchment response to a rainfall event. In addition, since this approach is proposed for hydrological applications, I am wondering whether it is much more convincing to feed those simulated scenarios into a hydrological model (event-based) for better validating this approach?

L280: Has the author developed/improved a novel method in this study. Please clarify it.

Figure 6 and Figure 7: these two figures are not informative. Please consider simplifying/redesigning these or moving them into the appendix.

---

## Referee Comment (RC2)

Peer review for "Hydrology and Earth System Sciences"

**Stochastic simulation of reference rainfall scenarios for hydrological applications using a universal multifractal approach**

Arun Ramanathan, Pierre-Antoine Versini, Daniel Schertzer, Remi Perrin, Lionel Sindt, and Ioulia Tchiguirinskaia.

Date Review: 02/21/2022

**Overall review comment:**

This paper presents the use of universal multifractal to generate ensembles of rainfall time series that recreates the Intensity (I), Duration (D), and Frequency (F) of rainfall time series, commonly used in the design of storm-water infrastructure. This paper may become an essential contribution to the literature body of stochastic simulations of rainfall time series. However, I found two pitfalls in the paper: (1) There is no clear definition of the research gap (including connections to previous works), and (2) Even though the paper assesses their methodology, the discussion about the results is almost non-existent. I hope my comments provide a road map to improve the important contribution done by the authors. Below there is a detailed description of my concerns.

**Major suggested comments:**

1. The research gap is vague. The authors must address the following questions. Why is it important to explore universal multifractals in rainfall datasets? Have studies been using this technique in rainfall datasets before? What are the challenges of using stochastic techniques and/or multifractals to reconstruct rainfall time series?

2. The ambiguity in the research gap is also reflected in the research objective: Line [57] states, "*The objective of this paper is to simulate region specific reference rainfall scenarios which could be used as realistic inputs (..) to hydrological models for optimally designing storm-water management infrastructures.*" This needs to be more specific. What method will be used? Will this be compared to a reference? Also, the objective elucidates "*hydrological models for optimally designing*"; however, the paper does not address optimal design at all. Readers may believe that the paper explores the use of universal multifractal for optimal designs for specific water-related infrastructure. However, the paper covers a general procedure for estimating ensembles of rainfall time-series, but it does not cover applications and consequences for engineering design.

3. There is no discussion section! The authors go from results to conclusions, omitting key discussions that will strengthen the overall contribution of their work. Here are some general pointers that I feel are relevant to find in the discussion section. 1) How are the results compared with the previous literature? Are the scaling parameters similar to previous studies? What are possible connections in the difference of scaling parameters between the three site studies? What are possible strategies to expand this methodology to spatial correlated rainfall datasets? What are potential limitations in the application of this

methodology? How feasible is this methodology to represent rainfall structure in large spatial scales? How can the spatial scale and limitations on this methodology affect the design of water-related infrastructure?

**Minor suggested comments:**

1. [75] Add reference for MeteoFrance
2. [110] Small sample size? Be more specific. What would be an ideal sample size?
3. [Figure] Figure 1 is not mentioned in the document
4. The term "resolution" is ambiguous throughout the text. Does it refer to time or space? Probably time.

---

## Author Comment (AC1)

**Response to Referee #1's comments on hess-2021-580:**

**General Comment:** *The manuscript presents a stochastic approach to generate an ensemble of reference rainfall scenarios (giving a desired rainfall amount P, rainfall duration D and return period T). This approach is based on multi-fractal theory, which is parsimonious and easy to apply. While this research topic is generally relevant for the readership of HESS, there is still a bit of issue which needs to be addressed. Please find the attached comments. Those comments should be addressed before considering publishing this article.*

> **Response:** We thank the referee for meticulously reviewing our manuscript and providing several constructive suggestions. We are especially grateful for the referee's positive feedback. In this document we provide our detailed response to the referee's supplementary comments and also mention how we plan to address these issues in a future version of this manuscript.

**Supplementary Comments:**

*Comment 1: L7: research gap is not clear.*

> **Reply:** It should be noted that this paper addresses the research gap between standard procedures for defining reference precipitation and the strong multi-scale intermittency of precipitation. It therefore proposes a procedure designed to tackle multi-scale intermittency head-on, based on extreme non-Gaussian statistics and scaling behaviour over a wide range of scales. The aim of this paper is that the baseline precipitation scenarios simulated by this procedure can be used as more realistic inputs into hydrological models for applications such as the optimal design of stormwater management infrastructure, including green roofs.
>
> We will incorporate the above explanation in the abstract to make the research gap that we address clearer right in the beginning of the paper.

*Comment 2: L21-23: it is not clear to me why 'uniform rainfall or … for such purposes' directly leads to the conclusion 'stochastic simulation of reference rainfall events is necessary'. At least one sentence (to discuss the lack of measurement data …) is missing.*

> **Reply:** We agree to improve the corresponding text. The sentence 'Uniform rainfall … for such purposes' will be shifted to the beginning of L57, whereas L21-24 will be replaced by

'Rainfall is quite commonly considered to be a stochastic variable due to the fact that rainfall process is complex and strongly dependent on initial conditions. There have been several studies/attempts to stochastically produce rainfall time series and space-time fields as listed here.'.

**Comment 3:** *L23: Does it refer to several types of single-site stochastic rainfall models or space-time models? Event-based simulation or continuous simulation? Please clarify.*

**Reply:** When we mention several model types in the beginning of L23 we refer to all these classes. To make this clearer, we will replace L23 by 'There have been several studies/attempts to stochastically produce rainfall time series and space-time fields as listed here. Simple point processes …'.

**Comment 4:** *L25: This paragraph is describing 'several types of stochastic rainfall models', but I am confused about why a model related to river discharge is described in detail. I get the point that one module in this 'Hydrogram model' is to generate hyetograms. If it is the case, I am wondering whether it is better to directly state that one of those types is the Monte Carlo method instead of the 'Hydrogram model'.*

**Reply:** We agree, and will replace L25-27 with ', and Monte Carlo method. All these four models are purely temporal.'

**Comment 5:** *A general comment on this whole paragraph (L38) & Table 1: The idea of creating this table looks good. However, more references would be needed. The latest one mentioned in this 'literature-based assessment' is in 2004, which is almost 20 years ago. In references for several model types for example in hybrid processes-based models, all references mentioned were published 20 years ago. Although those reference papers mentioned by the authors are very useful, it is necessary to update the reference list.*

**Reply:** We will add more recent references in this part and in Table. 1 following the referee's suggestion.

**Comment 6:** *L38: does it refer to spatial rainfall field or spatially averaged rainfall time series?*

**Reply:** For the sake of better clarity we will replace L38 with '1) Non-Homogeneity: Spatial Heterogeneity – rainfall is extremely variable with spatial location, especially at small

spatial scales and Temporal Heterogeneity or Intermittency – rainfall time series at a single spatial location is extremely variable with time, especially at small time scales'.

Since more than one query of the referee seems to be regarding space-time vs. time modelling, we will clarify that:

i. this dichotomy is not as strong as usual for multifractal models because a time multifractal can be seen as a time cut of a space-time multifractal

ii. the aim of the present study is focused over a fixed (and rather small) space extension such as a building roof

iii. the large scale deployment of rainfall-runoff management technologies would instead require space-time models, obtained with the help of new and rather limited developments (see (i)).

**Comment 7:** *L38: Does 'modelling approaches' refers to those modelling approaches for getting a realistic rainfall field for most hydrological applications or specifically for sizing storm-water management infrastructures?*

**Reply:** It refers to those modelling approaches (mentioned above from L23-36) for getting a realistic rainfall field for most hydrological applications (one such application is the designing of rain-water management infrastructures).

**Comment 8:** *L40: All these characteristics are '… makes the simulations physically relevant/realistic' while 'physically based' is specifically mentioned again in the second criteria 'Physically-based – the simulations rely on physical principles. What's the difference between these two? At least, the meaning of 'simulations rely on physical principles' in the second criteria need to be clarified.*

**Reply:** For improving clarity, we will replace 'physically relevant/realistic' by 'realistic' in L38. The term "physically-based" simply implies that these rainfall models represent the underlying process (at least abstractly) using physically meaningful parameters. This term is used in a slightly more generalized framework, because it is stochastic rather than deterministic, with fractional rather than integer derivatives.

**Comment 9:** *L56: The research gap is not ever mentioned until now. It would be much better to clearly state it in the manuscript.*

**Reply:** We will do this as mentioned in our reply to Comment 1 of the referee.

**Comment 10:** *L57: Single-site or multi-sites or spatial-temporal rainfall field or something else? please clarify.*

**Reply:** At present we simulate reference rainfall time series scenarios for different conurbations so single-site for three different sites (we will clarify this in L57). However, the advantage of the proposed simulation procedure is that since it is based on UM framework it can be easily extended in future to simulate spatio-temporal rainfall fields as well.

**Comment 11:** *L59-61: Ok. this seems to be the research aim, which gives me the impression that the authors are implying that: Using any existing stochastic rainfall models, it is NOT possible to "simulate reference rainfall ensembles characterized by P, D, T while exhibiting temporal variability and intermittency close to that of observed rainfall data". If my impression is correct, the authors were taking this as a research gap. However, this gap was not discussed or mentioned in the introduction section. I suggest revising the whole section and adding more essential information to it to make this introduction section easier to follow.*

**Reply:** Yes, as mentioned in our reply to Comment 1 of the referee we will modify the abstract.

**Comment 12:** *L78: resolution (space and time)?*

**Reply:** Just time.

**Comment 13:** *L120: are TM analysis and DTM analysis newly developed in this study? This is not clear to me.*

**Reply:** No, these are standard multifractal statistical analysis techniques and we have cited the corresponding papers in L120 and L136. However, this study uses a slightly modified iterative DTM procedure (also already existing) as explained in Appendix A. We felt it would be a bit more convenient for readers unacquainted with these analysis techniques if we added brief explanations here.

**Comment 14:** *L170: Is scale-symmetry able to be represented by other model types? From Table 1 in this manuscript, at least the Radar-based method can represent this, which can directly give us a space-time field. Why choose the multifractal theory instead of a radar-based*

*method (see Pegram's paper as mentioned by the authors)? It will be helpful to justify the choice and include this discussion (probably in the introduction section).*

**Reply:** The procedure proposed here needs only observational rainfall time series (not very data demanding) and is computationally simpler and parsimonious compared to the Radar-based bead method of Pegram. The current procedure can also be directly extended to obtain space-time fields as well. Furthermore, the idea of space-time complexity in the UM framework is somewhat more generalized than it is in the Radar-based bead model (spatial complexity and temporal complexity are dealt with separately rather than together). We will add a brief discussion about the Radar-based bead model and why we prefer the UM cascade model over it in the introduction section.

***Comment 15:*** *L172: The authors mention that these types of models can be considered as a bridge between purely statistical and purely physical models. I am wondering whether these features have been in this cascade model used in this study: seasonality (or different types of storm event).*

**Reply:** As mentioned in L168-171, this statement was made in the context of the UM cascade model using the physical concepts of energy transfer from large scales to small scales by random breakup of eddies to abstractly represent atmospheric processes underlying rainfall production. Statistical analysis of observed rainfall in the three conurbations chosen by this study did not display any significant seasonality, but there was a clear evidence of a strong synoptic maximum with corresponding changes in scaling behaviour. This required some elaboration of the UM cascade process to guarantee good agreement between observed and simulated rainfall over the full range of time scales, thereby reproducing well the statistics of different storm types (either convective or stratiform).

***Comment 16:*** *L176: It seems that the study presented here is to simulate temporal rainfall as the author mentioned in the abstract, but 'rainfall fields' convey a message that a time varying spatial field is generated. Not entirely sure if this word is widely used when time series is generated. If not, please change this word.*

**Reply:** We used the term field in a more generalized context since a time series can be considered as a one-dimensional field. However, for the sake of better clarity we will use 'rainfall time series' instead as suggested by the referee.

*Comment 17: L180: Does it always refer to temporal resolution?*

**Reply:** Yes.

*Comment 18: L225: Ok, I can see that these metrics might be useful. Besides these, I am wondering whether the autocorrelation of simulated events is well reproduced. This feature plays a critical role in dominating a catchment response to a rainfall event. In addition, since this approach is proposed for hydrological applications, I am wondering whether it is much more convincing to feed those simulated scenarios into a hydrological model (event-based) for better validating this approach?*

**Reply:** The Codimension Comparison Metric (CCM) already takes care of this issue and more. Autocorrelation or its inverse Fourier transform i.e. spectral density are generally just second order statistics. Comparing the scaling moment function $K(q)$ for $q = 2$ of observed and simulated rainfall is the same as comparing their respective spectra and therefore their autocorrelation. The CCM compares $c(\gamma)$ instead of $K(q)$ however they are just the Legendre transforms of each other.

It is true that the proposed approach is for hydrological applications such as designing green roofs for rain-water management, however, we need observational data of not only rainfall but also discharge from the green roof to validate the entire hydro-meteorological modelling approach. We are still working on setting up experimental green roof prototypes and monitoring protocols for this purpose, and the referee's query can only be addressed via a separate publication in the future.

*Comment 19: L280: Has the author developed/improved a novel method in this study. Please clarify it.*

**Reply:** Yes (as mentioned in our reply to referee's comments 1 and 11). Even though several earlier studies have attempted to simulate rainfall using a Universal Multifractal (UM) approach, we are unaware of UM-based studies that have proposed procedures to simulate reference rainfall scenarios. We will add this explanation in the beginning of L280.

***Comment 20:*** *Figure 6 and Figure 7: these two figures are not informative. Please consider simplifying/redesigning these or moving them into the appendix*

**Reply:** We feel the referee is referring to Figures 7 and 8. If so, we agree and will move them into the appendix.

---

## Author Response (AR1)

**Answers to both 1st and 2nd Referee's comments:**

**Answers to 1st Referee's comments:**

**General Comment:** *The manuscript presents a stochastic approach to generate an ensemble of reference rainfall scenarios (giving a desired rainfall amount P, rainfall duration D and return period T). This approach is based on multi-fractal theory, which is parsimonious and easy to apply. While this research topic is generally relevant for the readership of HESS, there is still a bit of issue which needs to be addressed. Please find the attached comments. Those comments should be addressed before considering publishing this article.*

> **Response:** We thank the referee for meticulously reviewing our manuscript and providing several constructive suggestions. We are especially grateful for the referee's positive feedback. In this document we provide our detailed response to the referee's supplementary comments and also mention how we have addressed these issues in the revised version of this manuscript.

**Supplementary Comments:**

***Comment 1:*** *L7: research gap is not clear.*

> **Reply:** It should be noted that this paper addresses the research gap between standard procedures for defining reference precipitation and the strong multi-scale intermittency of precipitation. It therefore proposes a procedure designed to tackle multi-scale intermittency head-on, based on extreme non-Gaussian statistics and scaling behaviour over a wide range of scales. The aim of this paper is that the baseline precipitation scenarios simulated by this procedure can be used as more realistic inputs into hydrological models for applications such as the optimal design of storm-water management infrastructure, including green roofs.
>
> We have added the above explanation in the abstract (L7) of the revised manuscript to make the research gap that we address clearer right at the beginning of the paper.

***Comment 2:*** *L21-23: it is not clear to me why 'uniform rainfall or … for such purposes' directly leads to the conclusion 'stochastic simulation of reference rainfall events is necessary'. At least one sentence (to discuss the lack of measurement data …) is missing.*

**Reply:** We agree and have improved the corresponding text in the revised manuscript. The sentence 'Uniform rainfall … for such purposes' has been shifted to L92, whereas L32 has been replaced by 'Rainfall is quite commonly considered to be a stochastic variable due to the fact that rainfall process is complex and strongly dependent on initial conditions.'. We have also added a sentence (L34) mentioning the lack of high-resolution observational data following the referee's suggestion.

*Comment 3:* *L23: Does it refer to several types of single-site stochastic rainfall models or space-time models? Event-based simulation or continuous simulation? Please clarify.*

**Reply:** When we mention several model types we refer to all these classes. To make this clearer, we have replaced this by 'There have been several studies/attempts to stochastically produce rainfall time series and space-time fields as listed here: Simple point processes …' in the revised manuscript (L36).

*Comment 4:* *L25: This paragraph is describing 'several types of stochastic rainfall models', but I am confused about why a model related to river discharge is described in detail. I get the point that one module in this 'Hydrogram model' is to generate hyetograms. If it is the case, I am wondering whether it is better to directly state that one of those types is the Monte Carlo method instead of the 'Hydrogram model'.*

**Reply:** We agree, and have replaced L39 with ', and Monte Carlo method… All these four models are purely temporal' in the revised manuscript.

*Comment 5:* *A general comment on this whole paragraph (L38) & Table 1: The idea of creating this table looks good. However, more references would be needed. The latest one mentioned in this 'literature-based assessment' is in 2004, which is almost 20 years ago. In references for several model types for example in hybrid processes-based models, all references mentioned were published 20 years ago. Although those reference papers mentioned by the authors are very useful, it is necessary to update the reference list.*

**Reply:** In the revised manuscript we have added more recent references in this part (L37-throughtout the paragraph) and in Table. 1 following the referee's suggestion.

*Comment 6:* *L38: does it refer to spatial rainfall field or spatially averaged rainfall time series?*

**Reply:** For the sake of better clarity we have replaced this with '1) Heterogeneity: Spatial Heterogeneity – rainfall is extremely variable with spatial location, especially at small spatial scales and Temporal Heterogeneity or Intermittency – rainfall time series at a single spatial location is extremely variable with time, especially at small time scales' in the revised manuscript (L56).

Since more than one query of the referee seems to be regarding space-time vs. time modelling, we have clarified (L95 of revised manuscript) that:

   i.    this dichotomy is not as strong as usual for multifractal models because a time multifractal can be seen as a time cut of a space-time multifractal

   ii.    the aim of the present study is focused over a fixed (and rather small) space extension such as a building roof

   iii.    the large scale deployment of rainfall-runoff management technologies would instead require space-time models, obtained with the help of new and rather limited developments (see (i)).

***Comment 7:*** *L38: Does 'modelling approaches' refers to those modelling approaches for getting a realistic rainfall field for most hydrological applications or specifically for sizing storm-water management infrastructures?*

**Reply:** It refers to those modelling approaches (mentioned in the previous paragraph) for getting a realistic rainfall field for most hydrological applications (one such application is the designing of rain-water management infrastructures). We have made this clearer in L54 of the revised manuscript.

***Comment 8:*** *L40: All these characteristics are '… makes the simulations physically relevant/realistic' while 'physically based' is specifically mentioned again in the second criteria 'Physically-based – the simulations rely on physical principles. What's the difference between these two? At least, the meaning of 'simulations rely on physical principles' in the second criteria need to be clarified.*

**Reply:** For improving clarity, we have replaced 'physically relevant/realistic' by 'realistic' in L55. The term "physically-based" simply implies that these rainfall models represent the underlying process (at least abstractly) using physically meaningful parameters. This term is used in a slightly more generalized framework, because it is stochastic rather than

deterministic, with fractional rather than integer derivatives. We have added this explanation in L59 of the revised manuscript.

*Comment 9: L56: The research gap is not ever mentioned until now. It would be much better to clearly state it in the manuscript.*

**Reply:** We have done this as mentioned in our reply to Comment 1 of the referee.

*Comment 10: L57: Single-site or multi-sites or spatial-temporal rainfall field or something else? please clarify.*

**Reply:** At present we simulate reference rainfall time series scenarios for different conurbations so single-site for three different sites (we have clarified this in L90). However, the advantage of the proposed simulation procedure is that since it is based on UM framework it can be easily extended in future to simulate spatio-temporal rainfall fields as well. This is mentioned in L98 of the revised manuscript.

*Comment 11: L59-61: Ok. this seems to be the research aim, which gives me the impression that the authors are implying that: Using any existing stochastic rainfall models, it is NOT possible to "simulate reference rainfall ensembles characterized by P, D, T while exhibiting temporal variability and intermittency close to that of observed rainfall data". If my impression is correct, the authors were taking this as a research gap. However, this gap was not discussed or mentioned in the introduction section. I suggest revising the whole section and adding more essential information to it to make this introduction section easier to follow.*

**Reply:** Yes, as mentioned in our reply to Comment 1 of the referee we have modified the abstract (L7) and this part (L81) in the revised manuscript.

*Comment 12: L78: resolution (space and time)?*

**Reply:** Just time, we have mentioned this clearly in Table. 3 and L116 of the revised manuscript.

*Comment 13: L120: are TM analysis and DTM analysis newly developed in this study? This is not clear to me.*

**Reply:** No, these are standard multifractal statistical analysis techniques and we have cited the corresponding papers in L163 and L179 of the revised manuscript. However, this study

uses a slightly modified iterative DTM procedure (also already existing) as explained in Appendix A. We felt it would be a bit more convenient for readers unacquainted with these analysis techniques if we added brief explanations here.

*Comment 14: L170: Is scale-symmetry able to be represented by other model types? From Table 1 in this manuscript, at least the Radar-based method can represent this, which can directly give us a space-time field. Why choose the multifractal theory instead of a radar-based method (see Pegram's paper as mentioned by the authors)? It will be helpful to justify the choice and include this discussion (probably in the introduction section).*

**Reply:** The procedure proposed here needs only observational rainfall time series (not very data demanding) and is computationally simpler and parsimonious compared to the Radar-based bead method of Pegram. The current procedure can also be directly extended to obtain space-time fields as well. Furthermore, the idea of space-time complexity in the UM framework is somewhat more generalized than it is in the Radar-based bead model (spatial complexity and temporal complexity are dealt with separately rather than together). We have added a brief discussion about the Radar-based bead model and why we prefer the UM cascade model over it in the introduction section (L76) of the revised manuscript.

*Comment 15: L172: The authors mention that these types of models can be considered as a bridge between purely statistical and purely physical models. I am wondering whether these features have been in this cascade model used in this study: seasonality (or different types of storm event).*

**Reply:** As mentioned in L219-222 of the revised manuscript, this statement was made in the context of the UM cascade model using the physical concepts of energy transfer from large scales to small scales by random breakup of eddies to abstractly represent atmospheric processes underlying rainfall production.

Although multifractal (statistical) analysis of observed rainfall in the three conurbations chosen by this study do not display any significant seasonality (as there is no scaling break around a few months time scale), there is a clear evidence of a strong synoptic maximum (indicated by a scaling break around few weeks time scale) with corresponding changes in scaling behaviour. It is worth noting that this aforementioned

absence of seasonality in multifractal characteristics means that the low frequency scaling regime's UM parameters are sufficient to represent seasonal variability (in cumulative precipitations), whereas together with the high frequency scaling regime's UM parameters they are sufficient for reproducing well the statistics of different storm types (either convective or stratiform). This requires some elaboration of the UM cascade process to guarantee good agreement between observed and simulated rainfall over the full range of time scales. This explanation has been added to L208 of the revised manuscript.

Furthermore, we have tried to compare seasonality (in a simplistic manner) in the simulated and observed rainfall as discussed in L355 of the revised manuscript.

***Comment 16:*** *L176: It seems that the study presented here is to simulate temporal rainfall as the author mentioned in the abstract, but 'rainfall fields' convey a message that a time varying spatial field is generated. Not entirely sure if this word is widely used when time series is generated. If not, please change this word.*

**Reply:** We used the term field in a more generalized context since a time series can be considered as a one-dimensional field. However, for the sake of better clarity we have used 'rainfall time series' instead as suggested by the referee in the revised manuscript (L227 and several other places).

***Comment 17:*** *L180: Does it always refer to temporal resolution?*

**Reply:** Yes, we have explicitly stated this in the revised manuscript (L231, L233).

***Comment 18:*** *L225: Ok, I can see that these metrics might be useful. Besides these, I am wondering whether the autocorrelation of simulated events is well reproduced. This feature plays a critical role in dominating a catchment response to a rainfall event. In addition, since this approach is proposed for hydrological applications, I am wondering whether it is much more convincing to feed those simulated scenarios into a hydrological model (event-based) for better validating this approach?*

**Reply:** The Codimension Comparison Metric (CCM) already takes care of this issue and more. Autocorrelation or its inverse Fourier transform i.e. spectral density are generally just second order statistics. Comparing the scaling moment function $K(q)$ for $q = 2$ of observed and simulated rainfall is the same as comparing their respective spectra and therefore their autocorrelation. The CCM compares $c(\gamma)$ instead of $K(q)$ however they

are just the Legendre transforms of each other. We have added this explanation in L330 of the revised manuscript.

It is true that the proposed approach is for hydrological applications such as designing green roofs for rain-water management, however, we need observational data of not only rainfall but also discharge from the green roof to validate the entire hydro-meteorological modelling approach. We are still working on setting up experimental green roof prototypes and monitoring protocols for this purpose, and the referee's query can only be addressed via a separate publication in future. We have added this part in the conclusions section of the revised manuscript (L379).

**Comment 19:** *L280: Has the author developed/improved a novel method in this study. Please clarify it.*

**Reply:** Yes (as mentioned in our reply to referee's comments 1 and 11). Even though several earlier studies have attempted to simulate rainfall using a Universal Multifractal (UM) approach, we are unaware of UM-based studies that have proposed procedures to simulate reference rainfall scenarios. We have added this explanation in the beginning of L363.

**Comment 20:** *Figure 6 and Figure 7: these two figures are not informative. Please consider simplifying/redesigning these or moving them into the appendix*

**Reply:** We feel the referee is referring to Figures 7 and 8. If so, we agree and have moved them into the appendix.

**Answers to 2nd Referee's comments:**

**General Comment:** *This paper presents the use of universal multifractal to generate ensembles of rainfall time series that recreates the Intensity (I), Duration (D), and Frequency (F) of rainfall time series, commonly used in the design of storm-water infrastructure. This paper may become an essential contribution to the literature body of stochastic simulations of rainfall time series. However, I found two pitfalls in the paper: 1) There is no clear definition of the research gap (including connections to previous works), and (2) Even though the paper assesses their methodology, the discussion about the results is almost non-existent. I hope my*

*comments provide a road map to improve the important contribution done by the authors. See the attached file for a detailed description of my concerns.*

**Response:** We thank the referee for reviewing our manuscript and providing several constructive suggestions. We are especially grateful for the positive feedback. In this document we provide our detailed response to the detailed description of the reviewer's comments and also mention how we have addressed these issues (especially regarding research gap definition and result discussion) in the revised version of this manuscript.

**Major Suggested Comments:**

*Comment 1: The research gap is vague. The authors must address the following questions. Why is it important to explore universal multifractals in rainfall datasets? Have studies been using this technique in rainfall datasets before? What are the challenges of using stochastic techniques and/or multifractals to reconstruct rainfall time series?*

**Reply:** Following the referee's remarks and suggestions, we have specified the research gaps, which are of three kinds:

- a general discrepancy between standard procedures for defining reference precipitation and the strong multiscale intermittency of precipitation.
- missing procedure to adapt multifractal precipitation modelling to given partial statistical references.
- missing procedure to assess the accuracy of the method.

The corresponding challenges addressed in this paper are:

- to tackle multiscale intermittency head-on, based on extreme non-Gaussian statistics and scaling behaviour over two subranges of time scales, due to the finite size of the earth. This requires a given adaptation of the multifractal modelling procedure.
- to define a renormalizing procedure for the multifractal model to make the simulations fit with these partial statistical references.
- to define multiscale metrics to assess distance between (closeness of) two time series (observed and simulated) across time scales.

This will enable us to provide baseline precipitation scenarios that can be used as realistic inputs into hydrological models for applications such as the optimal design of storm-water management infrastructure, especially green roofs.

We have incorporated the above explanation in the abstract (L7), and introduction (L81) to make the research gap addressed clearer right at the start of the paper.

***Comment 2:*** *The ambiguity in the research gap is also reflected in the research objective: Line [57] states, "The objective of this paper is to simulate region specific reference rainfall scenarios which could be used as realistic inputs (..) to hydrological models for optimally designing storm-water management infrastructures." This needs to be more specific. What method will be used? Will this be compared to a reference? Also, the objective elucidates "hydrological models for optimally designing"; however, the paper does not address optimal design at all. Readers may believe that the paper explores the use of universal multifractal for optimal designs for specific water-related infrastructure. However, the paper covers a general procedure for estimating ensembles of rainfall time-series, but it does not cover applications and consequences for engineering design.*

**Reply:** We hope that the above clarification (Reply to comment 1) will dissipate any ambiguity on the goal of the paper. Moreover, we have further clarified that designing optimal storm-water management infrastructures (such as green roofs) is not an objective of this paper but a scope for applications of the reference rainfall simulation procedure (which makes use of the UM framework to properly take into account the temporal variability of rainfall) proposed here. This explanation has been added to L81-90 of the revised manuscript.

We would also like to mention that we are currently working to set-up experimental green roof prototypes (designed based on reference rainfall scenarios simulated using the method proposed in this manuscript) and establishing monitoring protocols, therefore the applicability of the present results is beyond a good wish and referee's comment will be further addressed via a separate publication in future. This explanation has been added to L379 of the revised manuscript.

***Comment 3.1:*** *There is no discussion section! The authors go from results to conclusions, omitting key discussions that will strengthen the overall contribution of their work. Here are*

*some general pointers that I feel are relevant to find in the discussion section. 1) How are the results compared with the previous literature? Are the scaling parameters similar to previous studies? What are possible connections in the difference of scaling parameters between the three site studies?*

> **Reply:** We agree and have inserted a specific discussion section based on result discussion already present in the paper. We have added a discussion part in Line [336] addressing these queries.

***Comment 3.2:** What are possible strategies to expand this methodology to spatial correlated rainfall datasets? What are potential limitations in the application of this methodology? How feasible is this methodology to represent rainfall structure in large spatial scales? How can the spatial scale and limitations on this methodology affect the design of water-related infrastructure?*

> **Reply:** In the context of these queries regarding space-time vs. time modelling, we have clarified in L94 of the revised manuscript that:
>
> i.      this dichotomy is not as strong as usual for multifractal models because a time multifractal can be seen as a time cut of a space-time multifractal
>
> ii.      the aim of the present study (which mainly focusses on temporal scales) is focused over a fixed (and rather small) space extension such as a building roof
>
> iii.      the large scale deployment of rainfall-runoff management technologies/infrastructures would instead require space-time models (and space-time rainfall datasets), obtained with the help of new and rather limited developments (see (i)).

**Minor Suggested Comments:**

***Comment 1:** [75] Add reference for MeteoFrance.*

> **Reply:** We have added it in the revised manuscript (L118).

***Comment 2:** [110] Small sample size? Be more specific. What would be an ideal sample size?*

> **Reply:** In line [147] we have mentioned that a single sample is used, so here the small sample (effective) dimension is 1. Larger the sample size, better will be the estimate of

spectral slope (better straight line fit). But increasing sample size with a fixed dataset length means that with more samples the length of each sample is smaller, implying that there is a reduction in the largest scale considered. This may in turn lead to a difference in multifractal characteristics. The TM analysis, on the other hand, does not have this disadvantage and the straight line fits are reasonably good and not too dependent on the number of samples. Therefore, TM analysis is simply more preferable/relevant compared to spectral analysis or the question of how many samples will be ideal when using spectral analysis. We have added this explanation in L151 of the revised manuscript.

**Comment 3:** *[Figure] Figure 1 is not mentioned in the document*

**Reply:** It is mentioned in Line [69] of the revised manuscript.

**Comment 4:** *The term "resolution" is ambiguous throughout the text. Does it refer to time or space? Probably time.*

**Reply:** The interest of the term "resolution" is that it is dimensionless because it is precisely defined by the ratio of the outer scale by the inner scale. As such, it is valid for time and/or space, but presently used for time. This has been specified in the revised manuscript to avoid any ambiguity.

---

## Referee Report (RR1)

**General assessment:**
This is a rather low quality manuscript. Its two main weaknesses are 1) lack of novelty and 2) poor writing/structure. The first relates to the fact that UM cascade have been used to analyze/simulate rainfall time series for a long time, including extremes. Therefore, the contribution of the present paper remains unclear. In terms of writing, I would say that the paper is lengthy and overly complicated. The text contains many digressions on other, irrelevant issues that are outside the scope of the paper and distract the reader from the essential. Also, there is a lot of jargon, which makes the research sound more complicated than it really is. At the same time, lots of important practical details about the actual implementation of the cascade are missing, which makes it impossible to reproduce the work.

**My recommendation:** major revisions.

**Major comments:**

a) The abstract does not state/summarize the most important results. It is too long and misrepresents the scope of the paper.

> Suggestion: rewrite the entire abstract. Describe what this paper is about, highlighting the novelty and contribution. Be concise and clearly mention the main results.

b) The Introduction is too broad and contains irrelevant information. This paper is about the simulation of rainfall time series using discrete UM cascades. Therefore, I do not see any need to dwell on spatial models and space-time models. The part about the 8 characteristics of rainfall fields on lines 53-67 is not necessary for the understanding of the paper, and so are Table 1 and Fig 1. Most of these aspects never come back in the results part of the paper.

> Suggestion: tighten the scope of the introduction/methods by focusing on time series only. Instead of wandering off topic, include more relevant background information about the current weaknesses/strengths of rainfall time series models, including their ability to reproduce extremes and IDFs. Highlight what the knowledge gap(s) is/are and how the methods proposed in this paper address it/them.

c) Figure 1 and Table 1 are not necessary for the comprehension of the paper. The computational complexity never comes back and none of the other methods are implemented/used.

> Suggestion: remove/shorten them or consider adding other methods to compare against.

d) Fig 1 and Table 1 are deeply misleading. Within a given category, many different implementations/flavors have been proposed. The complexity and number of parameters vary a lot depending on which publication you consider.

> Suggestion: to make the comparison fair, you should refer to specific papers (e.g., authors + year + name of method) or give a range of values for multiple publications.

e) The random cascade model implemented in this paper uses 4 parameters and not 2 as claimed in Table 1. Therefore, it is not objectively more parsimonious than many of the other methods mentioned in the introduction. According to your own definition in Fig 1, the model would not be labeled as highly parsimonious.

> Suggestion: do not label models as highly parsimonious, etc. Focus on explaining the differences in approach, and how much of the original complexity can be reproduced with a

given set of parameters. Depending on the application, different characteristics will be important, such as extremes, mean, variance, autocorrelation, intermittency etc. Clearly explain which characteristics are the most important to you.

Note: actually, the number of parameters is 5, because you also need to count the scale break (which needs to be estimated from the data).

f) There is crucial information missing about how the cascade models are implemented, and how the time series are generated. Because of this, the research is impossible to reproduce.

Suggestion: restructure section 4. Consider creating more sub-sections in 4.1 to explain the different parts, from the simulation itself (using the Lévy random variables) to the renormalization. Provide a step-by-step description and mention the software packages/tools used. If possible, provide documented example codes.

g) The Results/Discussion part is too short and too shallow. The outcomes need to be discussed in more details. The scores are not enough to understand/interpret the results.

Suggestion: extend the Discussion part. Include more diagnostic plots and critically discuss the pros/cons. If possible, compare the outcomes to what is possible to achieve with another of the mentioned simulation techniques (not UM based).

h) Be more critical with respect to obtained results. While reading the paper, I got the impression that the authors were very quick at praising the UM cascade model and how amazing it is. However, UM cascade also come with limitations and the whole approach relies on some pretty strong assumptions which need to be discussed.

Suggestion: objectively report on what the method can/cannot do and critically discuss the assumptions it relies on.

i) explicitly state what you actually mean by seasonality. Different characteristics of the precipitation process may have different seasonal patterns. For example the wet/dry spell lengths, the average precipitation amounts or the extremes. In addition, you don't actually need the UM framework to assess seasonality.

Suggestion: clearly define what seasonality means in the context of this paper and use traditional metrics such as the coefficient of variation (or related) to quantify the observed/simulated seasonality. Check whether the UM cascade can reproduce these quantities.

j) lots of self-referencing: Out of the 71 references, at least 27 refer to work done by people in the same group as the co-authors. This is a lot and could be qualified as excessive self-citation.

Suggestion: check whether all these self-references are really needed.

**Minor comments:**

- Throughout the paper: avoid using too many parentheses at the end of your sentences. This gets annoying very quickly and is bad writing. Just add a new sentence or consider combining the two parts using a comma.

- The TM and DTM methods have already been explained in great detail in other studies. You could save space by not repeating the theory and referring to the relevant papers.

- ll.46-47: *"[…] however they do make some non-physical simplifying assumptions […]"*

    Which ones?

- ll.53-67: when mentioning the 8 properties, you should better distinguish actual properties (as seen in observations) from model properties.

- l.60: *"[…] for instance fields are not presumed to be additive"*

    Please explain what you mean by this. Are you referring to additive errors?

- ll.63-64: *"[…] extreme rainfall values are more frequent than usual resulting into strongly non-Gaussian statistics."*

    Nonsense. By definition, extreme values are less frequent than usual ones. Just say that the distributions are positively skewed, with long right tails.

- l.145 Equation 2: Why not give the general expression with the H?

- l.151: *"Larger the sample size, better will"*

    Gibberish

- ll.184-185: *"Generally, this could be due to two different issues: […]"*

    The most plausible issue should also be mentioned here: that the data are not really multi-fractal. In other words: the assumption itself should be questioned (on top of how the parameters are estimated).

- ll.207-208: "These low values of MCI justify the aforementioned selection procedure"

    Well, maybe. But without context, this number does not mean much. What is an acceptable value?

- ll.221-222: *"[…] a property respected even by the Navier-Stokes equation used by state-of-the-art NWP models for operational forecasting"*

    Please add a reference here.

- l.223: "[…] can be considered as a bridge between purely statistical and purely physical models"

    Nonsense. A bridge is what you use to cross from one side to another. Here, you just have a method that combines the properties of both worlds. But that does not make it a bridge and does not tell us how to go from the physical to the statistical world.

- l.236: "[…] and suitable amplitude [...]"

make this part more explicit by stating exactly how the Lévy variable is simulated. See major comment (f).

- l.351: "[…] it can be seen that they are somewhat similar to another."

  Too vague. Provide the absolute and relative differences. Actually, one could make the point that since the UM parameters for different sites with different rainfall properties are very similar, they do not really offer a great physical interpretation. Otherwise, one would be able to see the differences just by looking at the parameter values. Is this because small differences in parameter values can have large differences in terms of patterns? Please elaborate.

- ll.360-364: "[…] thereby confirming that the simulations have reasonably realistic seasonality features."

  I don't think that you have presented enough evidence to conclude this. The simple, subjective comparison with some ratios close to 0 is very sketchy and some more in-depth analyses and diagnostic plots are necessary to convince me of the realism of seasonal features in the simulations.

- l.374: […] " statistically realistic reference rainfall ensembles"

- ll.377-378: "[…] seems to be the most reliable comparison metric."

  Reliable is a strange word in this context. Did you mean robust? Or adequate?

- ll.388-391: This is a strange way to conclude a paper. This paragraph would fit better in the Introduction, to justify the UM cascade model.

---

## Author Response (AR2)

**Response to the editor:**

We confess that we were extremely surprised that the third referee, who only participated in the last round, seems to have ignored the previous rounds. In particular, he totally ignored our detailed responses on the novelty of the paper and the corresponding changes in the manuscript. Moreover, no dissatisfaction from the first two referees was communicated to us. As this was the only main weakness in the paper content claimed by the referee, we do not understand why a major revision was decided. Indeed, the second main weakness claim was limited to the quality of the writing.

Despite these objections, we have carefully analysed, commented on and taken into account the referee's numerous comments, which focused mainly on the presentation of the manuscript.

**Response to 3rd Anonymous Referee's assessment/recommendation/comments/suggestions:**

**General assessment:**

*This is a rather low quality manuscript. Its two main weaknesses are 1) lack of novelty and 2) poor writing/structure. The first relates to the fact that UM cascade have been used to analyze/simulate rainfall time series for a long time, including extremes. Therefore, the contribution of the present paper remains unclear. In terms of writing, I would say that the paper is lengthy and overly complicated. The text contains many digressions on other, irrelevant issues that are outside the scope of the paper and distract the reader from the essential.*

*Also, there is a lot of jargon, which makes the research sound more complicated than it really is.*

*At the same time, lots of important practical details about the actual implementation of the cascade are missing, which makes it impossible to reproduce the work.*

**Response:** We are grateful to the referee for reviewing our manuscript in detail and providing numerous comments and suggestions. However, the referee seems to have ignored the previous rounds. In particular, he totally ignored our detailed responses on the novelty of the paper and the corresponding changes in the manuscript. Following questions of the two first referees, we sharpened out that we addressed three kinds of knowledge gaps:

- a general discrepancy between standard procedures for defining reference precipitation and the strong multiscale intermittency of precipitation.
- missing procedure to adapt multifractal precipitation modelling to given partial statistical references.
- missing procedure to assess the accuracy of the method.

with the corresponding challenges:

- to tackle multiscale intermittency head-on, based on extreme non-Gaussian statistics and scaling behaviour over two subranges of time scales, due to the finite size of the earth. This requires a given adaptation of the multifractal modelling procedure.
- to define a renormalizing procedure for the multifractal model to make the simulations fit with these partial statistical references.
- to define multiscale metrics to assess distance between (closeness of) two time series (observed and simulated) across time scales.

In short, one should not confuse the UM cascades used by earlier studies to simulate rainfall (without any reference constraints), and the UM cascades being used here to simulate reference rainfall scenarios (with the constraint of certain durations, return periods and intensities) that could be used to optimise the design of certain urban storm-water management infrastructure.

The second weakness claim is limited to our writing quality, not to the content of the paper. Unfortunately, this is a somewhat subjective point-of-view and is partly due to our text edits based on suggestions from other referees. However, we have reviewed the

whole text, in particular to limit the use of multifractal jargon to what is absolutely essential for this paper.

We respectfully disagree with the referee's comment on the work being impossible to reproduce, since it is fully based on the discrete UM cascades that have been explained in great detail in several earlier studies, some of them cited here.

In this document we provide our detailed response to the referee's comments/suggestions and also mention how we have addressed some of these queries in the revised version of this manuscript.

**Recommendation:** *major revisions.*

**Response**: We consider that the term "major" is not supported by the above responses, in particular those addressed to the first two referees we have recalled. Most of the current comments and corresponding suggestions, several of which correspond to the literature review based Table and Figure in the introductory part, simply call for some text edits! It is worth to note that 7 out of the 10 "major" comments where rather focused on the presentation of the results, not the on the results themselves.

**Major Comments:**

**_Comment a:_** *The abstract does not state/summarize the most important results. It is too long and misrepresents the scope of the paper.*

**_Suggestion a:_** *rewrite the entire abstract. Describe what this paper is about, highlighting the novelty and contribution. Be concise and clearly mention the main results.*

**Reply:** This revised abstract, although a bit long, was actually a result of modifications considering comments from referees #1 and #2. It lists the 3 research gaps and the corresponding 3 contributions that we recalled above, they highlight the important novelty of the paper. Therefore, we respectfully disagree with the referee.

**_Comment b:_** *The Introduction is too broad and contains irrelevant information. This paper is about the simulation of rainfall time series using discrete UM cascades. Therefore, I do not see any need to dwell on spatial models and space-time models. The part about the 8*

*characteristics of rainfall fields on lines 53-67 is not necessary for the understanding of the paper, and so are Table 1 and Fig 1. Most of these aspects never come back in the results part of the paper.*

**Suggestion b:** *tighten the scope of the introduction/methods by focusing on time series only. Instead of wandering off topic, include more relevant background information about the current weaknesses/strengths of rainfall time series models, including their ability to reproduce extremes and IDFs. Highlight what the knowledge gap(s) is/are and how the methods proposed in this paper address it/them.*

> **Reply:** The discussion about space-time cascades was added in response to comments from referee #2. We partly agree with referee#3 that the last 2 characteristics - high parameter parsimony and low computational complexity – seem at first glance not essential for the understanding, but are nevertheless of prime importance for data analysis and stochastic simulations, and thus for the choice of methods.
>
> Anyway, the other characteristics (except space-time complexity) are in line with the referee's suggestion. The recommendation to include information about the weaknesses of current rainfall time series models in simulating extremes was in fact done earlier: the necessity to go well beyond Gaussian statistics has already been explained in L64.

**Comment c:** *Figure 1 and Table 1 are not necessary for the comprehension of the paper. The computational complexity never comes back and none of the other methods are implemented/used.*

**Suggestion c:** *remove/shorten them or consider adding other methods to compare against.*

> **Reply:** We partly agree with the referee, and as mentioned in our response to comment b, we mention parameter parsimony and computational complexity in Fig.1 and Table. 1 only as bare facts, e.g. 5 parameters over time periods ranging from a few minutes/hours to years. As the referee points out below, any attempt to compare with other methods easily becomes complex and/or partial. On the contrary, we briefly recall that for each scaling range UM enable to work with the minimal number of parameters (2) that is theoretically needed to obtain multiscaling, i.e. a nonlinear scaling function (Schertzer and Lovejoy, 1987, 1997).

***Comment d:*** *Fig 1 and Table 1 are deeply misleading. Within a given category, many different implementations/flavors have been proposed. The complexity and number of parameters vary a lot depending on which publication you consider.*

***Suggestion d:*** *to make the comparison fair, you should refer to specific papers (e.g., authors +year + name of method) or give a range of values for multiple publications.*

> **Reply:** This issue is resolved in the revised manuscript (see Table. 1, Fig. 1) as mentioned in our response to comments b and c, i.e. there can be no more room for misunderstanding or uncertainty.

***Comment e:*** *The random cascade model implemented in this paper uses 4 parameters and not 2 as claimed in Table 1. Therefore, it is not objectively more parsimonious than many of the other methods mentioned in the introduction. According to your own definition in Fig 1, the model would not be labeled as highly parsimonious.*

***Suggestion e:*** *do not label models as highly parsimonious, etc. Focus on explaining the differences in approach, and how much of the original complexity can be reproduced with a given set of parameters. Depending on the application, different characteristics will be important, such as extremes, mean, variance, autocorrelation, intermittency etc. Clearly explain which characteristics are the most important to you.*

***Note:*** *actually, the number of parameters is 5, because you also need to count the scale break (which needs to be estimated from the data).*

> **Reply:** The issue of parameter parsimony is resolved in the revised version as mentioned in our response to comments b, c and d, in particular we clarified, thanks to referee's remarks, that the minimal number of parameters is 2 per scaling sub-ranges.

***Comment f:*** *There is crucial information missing about how the cascade models are implemented, and how the time series are generated. Because of this, the research is impossible to reproduce.*

***Suggestion f:*** *restructure section 4. Consider creating more sub-sections in 4.1 to explain the different parts, from the simulation itself (using the Lévy random variables) to the renormalization. Provide a step-by-step description and mention the software packages/tools used. If possible, provide documented example codes.*

**Reply:** As mentioned in our response to the referee's general assessment, the simulation procedure is fully based on the discrete UM cascades that have been explained in great detail in several earlier studies, some of which have been cited here. Furthermore, Fig. 5 already shows the step-by-step algorithm used by the Python-based simulation code. Moreover, in line with the referee's demand to have a paper focused on the novelty, we do not feel we can give in more details in the present paper. However, the interested reader has all the means to reproduce the simulations.

**_Comment g:_** _The Results/Discussion part is too short and too shallow. The outcomes need to be discussed in more details. The scores are not enough to understand/interpret the results._

**_Suggestion g:_** _extend the Discussion part. Include more diagnostic plots and critically discuss the pros/cons. If possible, compare the outcomes to what is possible to achieve with another of the mentioned simulation techniques (not UM based)._

**Reply:** Our idea of defining the metrics was to make a quantitative, robust yet quick comparison of the simulations with observed datasets, and they seem quite adequate considering the objective of this manuscript. It should be noted that these metrics (MCM, SCM, CCM) are defined across scales, unlike the usual scores (such as RCM) which are limited to the estimation of a given scale. It is also worth noting that discussion of results in this paper is not restricted to section 5. Section 2 and 3 discuss some data analysis results, whereas section 4 discusses simulation results.

**_Comment h:_** _Be more critical with respect to obtained results. While reading the paper, I got the impression that the authors were very quick at praising the UM cascade model and how amazing it is. However, UM cascade also come with limitations and the whole approach relies on some pretty strong assumptions which need to be discussed._

**_Suggestion h:_** _objectively report on what the method can/cannot do and critically discuss the assumptions it relies on._

**Reply:** One main limitation in this paper is that of discrete UM cascades, they use integer scale-ratios which can be considered to be a non-physical assumption. We have already mentioned this in L232, L399. The method proposed here can only do what it was developed for i.e. simulating realistic reference rainfall scenarios to design storm-water management infrastructure. Simulating rainfall in real time and/or forecasting rain is not

the goal of this method. Furthermore, it cannot be used directly to simulate additional related variables such as temperature that could be relevant in the design of urban storm-water management devices including green roofs.

***Comment i:*** *explicitly state what you actually mean by seasonality. Different characteristics of the precipitation process may have different seasonal patterns. For example, the wet/dry spell lengths, the average precipitation amounts or the extremes. In addition, you don't actually need the UM framework to assess seasonality.*

***Suggestion i:*** *clearly define what seasonality means in the context of this paper and use traditional metrics such as the coefficient of variation (or related) to quantify the observed/simulated seasonality. Check whether the UM cascade can reproduce these quantities.*

> **Reply:** As mentioned in L366 - 374**,** we use the time gap between the maximum and minimum monthly average of cumulative precipitation as an indicator of seasonality. The *classical* UM framework does not address seasonality because it assumes a form of statistical stationarity. However, this framework can be generalised to include a given type of seasonality (Tchiguirinskaia et al. 2002). To keep the present paper as focused as possible, we only wanted to take into account a question of the referee #1 on possible biases of UM simulation vs. empirical data due to the difference of periodicity. This is why we use this simple indicator, which just assesses whether the time gap between the maximum and minimum monthly rainfall is similar for both observed and simulated rainfall. With respect to the traditional coefficient of variation, it had the advantage not to be limited to quasi-Gaussian/second order statistics. Again, to keep the paper as focused as possible, we do not feel we have to elaborate more.

***Comment j:*** *lots of self-referencing: Out of the 71 references, at least 27 refer to work done by people in the same group as the co-authors. This is a lot and could be qualified as excessive self-citation.*

***Suggestion j:*** *check whether all these self-references are really needed.*

> **Reply:** We have removed a few references that weren't too relevant in the revised manuscript.

**Minor Comments:**

***Comment 1:*** *Throughout the paper: avoid using too many parentheses at the end of your sentences. This gets annoying very quickly and is bad writing. Just add a new sentence or consider combining the two parts using a comma.*

**Reply:** We have reduced the usage of parenthesis in the revised version.

***Comment 2:*** *The TM and DTM methods have already been explained in great detail in other studies. You could save space by not repeating the theory and referring to the relevant papers.*

**Reply:** Although we agree, these parts are already not that space consuming. While the description of TM method is around 5-6 lines, that concerning basic DTM is around 5 lines.

***Comment 3:*** *ll.46-47: "[…] however they do make some non-physical simplifying assumptions […]": Which ones?*

**Reply:** L78-80 already discusses such a simplification in Radar-based bead models. Cell clusters and Modified turning band models both make Gaussian assumptions. We have added this later sentence in the revised manuscript.

***Comment 4:*** *ll.53-67: when mentioning the 8 properties, you should better distinguish actual properties (as seen in observations) from model properties.*

**Reply:** Since the UM model parameters correspond to data statistical estimators (as mentioned in L55) such a distinction is rather limited to the fact that the latter has uncertainties.

***Comment 5:*** *l.60: "[…] for instance fields are not presumed to be additive": Please explain what you mean by this. Are you referring to additive errors?*

**Reply**: Not at all, but to the fact that the underlying processes are presumably not additive, e.g. like a Gaussian or a Lévy process, but multiplicative. The former are linear, while the latter are strongly nonlinear**.** Therefore, we use the UM cascade models where the Levy distribution is used only for simulating the generator which is then exponentiated to obtain rainfall. Although this is already explained a bit in L239, we have made this clearer in the revised manuscript.

**_Comment 6:_** _ll.63-64: "[…] extreme rainfall values are more frequent than usual resulting into strongly non-Gaussian statistics.": Nonsense. By definition, extreme values are less frequent than usual ones. Just say that the distributions are positively skewed, with long right tails._

**Reply:** We meant extremes occur more frequently in fat-tailed distributions than in Gaussian distributions. So the comparison was obviously between the occurrence of extreme events in Gaussian and non-Gaussian distributions, not between extremes and usual events! We have made this clearer in the revised manuscript.

**_Comment 7:_** _l.145 Equation 2: Why not give the general expression with the H?_

**Reply:** Ok, we have given the generalized expression in the revised manuscript.

**_Comment 8:_** _l.151: "Larger the sample size, better will": Gibberish_

**Reply:** Unfortunately, it is unclear what the referee thinks is Gibberish here as the entire sentence is "Larger the sample size, better will be the estimate of spectral slope". If the issue is with how to get a larger sample, then we added the example that "Spectral slope obtained from a time series that is split into a number of smaller samples is more reliable than that obtained from the whole time series".

**_Comment 9:_** _ll.184-185: "Generally, this could be due to two different issues: […]": The most plausible issue should also be mentioned here: that the data are not really multifractal. In other words: the assumption itself should be questioned (on top of how the parameters are estimated)._

**Reply:** With finite samples, we can only estimate how much the observed field _could reasonably be_ multifractal, but not how much it _is really_ multifractal. This is achieved by assessing how closely the empirical statistical moments follow a scaling law for each moment order over a given range of resolutions. We think Figure. 3 already made this point very clear.

**_Comment 10:_** _ll.207-208: "These low values of MCI justify the aforementioned selection procedure": Well, maybe. But without context, this number does not mean much. What is an acceptable value?_

**Reply:** As shown in Eqs. 6, 7 the MCI here is totally dependent on $\alpha, C_1$. Since $0 \leq \alpha \leq 2$ and $0 \leq C_1 \leq 1$ (due to the assumption of a single sample), this implies that the maximum and minimum value of $\gamma_s$ are close to 1, 0 respectively. Therefore, it is rather straightforward to see that the maximum value of MCI is around 1 due to which the MCI values obtained in the text (0.03, 0.03, 0.04) are low and can be considered acceptable. We have added this explanation in the revised manuscript.

**Comment 11:** ll.221-222: "[...] a property respected even by the Navier-Stokes equation used by state-of-the-art NWP models for operational forecasting": Please add a reference here.

**Reply:** Ok.

**Comment 12:** l.223: "[...] can be considered as a bridge between purely statistical and purely physical models": Nonsense. A bridge is what you use to cross from one side to another. Here, you just have a method that combines the properties of both worlds. But that does not make it a bridge and does not tell us how to go from the physical to the statistical world.

**Reply:** These cascade models are based on Richardson's idea of energy transfer embodied in his 1922 Poem "Big whorls have little whorls Which feed on their velocity, And little whorls have lesser whorls And so on to viscosity." So the ideology of cascade models is firmly rooted in the so called physical world, while generating fields that have the right statistical properties. Therefore, these cascade models take us from the physical world to the statistical world due to which we see no issue in calling them a bridge between these two worlds. The importance of this type of bridge has gained recognition from the Nobel Committee for Physics, (Schertzer and Nicolis, 2022).

**Comment 13:** l.236: "[...] and suitable amplitude [...]": make this part more explicit by stating exactly how the Lévy variable is simulated. See major comment (f).

**Reply:** As mentioned in our response to comment (f) there are several studies that have already explained such simulation procedures in great detail, and we have already cited them in L237, L243.

**Comment 14:** l.351: "[...] it can be seen that they are somewhat similar to another.": Too vague. Provide the absolute and relative differences. Actually, one could make the point that since the UM parameters for different sites with different rainfall properties are very similar,

they do not really offer a great physical interpretation. Otherwise, one would be able to see the differences just by looking at the parameter values. Is this because small differences in parameter values can have large differences in terms of patterns? Please elaborate.

> **Reply:** Similarity of the parameter values confirms that rainfall at the three different locations have some common properties, e.g. intermittency. At the same time, small differences in parameter values can result in significant changes in the probability of occurrence of events exceeding a given threshold, therefore possible location dependent processes, for instance, different levels of intermittency. We have added this explanation in the revised manuscript.

***Comment 15:*** ll.360-364: "[…] thereby confirming that the simulations have reasonably realistic seasonality features.": I don't think that you have presented enough evidence to conclude this. The simple, subjective comparison with some ratios close to 0 is very sketchy and some more in-depth analyses and diagnostic plots are necessary to convince me of the realism of seasonal features in the simulations.

> **Reply:** As mentioned in our response to comment i, although this was in no way the primary objective of this manuscript. Indeed, we only answered to a question of referee #1 on possible biases related to periodicity. We suggested a very simple metric to have a first look to it. Let us underline that the maximum time gap in months will give a metric close to 1, while a value close to 0 suggests that the observed time gap and simulated time gap between maximum and minimum monthly rainfall is very similar.

***Comment 16:*** l.374: […] " statistically realistic reference rainfall ensembles"

> **Reply:** We feel that the simulations being physically and statistically realistic go hand in hand. The reason is that the UM Framework and its parameters, unlike those of simpler are physically meaningful (as already explained in L221 & L352), consequently they help to produce rainfall scenarios with the right statistics and probably the right physics.

***Comment 17:*** ll.377-378: "[…] seems to be the most reliable comparison metric.": Reliable is a strange word in this context. Did you mean robust? Or adequate?

**Reply:** We feel the word reliable is rather adequate here, given the fact that traditional metrics like the RCM seem too dependent on dataset sizes, therefore being unreliable for quantitatively comparing simulations with observations.

***Comment 18:*** ll.388-391: This is a strange way to conclude a paper. This paragraph would fit better in the Introduction, to justify the UM cascade model.

**Reply:** Ok, this line has been removed.

---

## Author Response (AR4)

**Response to Anonymous Referee #3's Comments:**

**General Comments:**

*This is the second time that I review this paper. Sadly, the authors chose to ignore most of my feedback. They have stated their reasons in their rebuttal. However, I do not find their arguments particularly convincing. Since the authors and I clearly disagree about the novelty, originality and quality of this work, I do not see any point in continuing this review and kindly ask the editor to invite another, impartial reviewer to replace me and continue the review process in the best, most objective way.*

**Reply:** We thank the referee for having agreed to review our manuscript again and for acknowledging that our rebuttal was in fact rather factual. We have therefore not ignored the arbitrator's comments, particularly when we counter-argued them. We can only regret that the referee limited themselves to stating that we disagree and resign without specifying what those disagreements actually are and we are unfortunately at a loss to respond any further.

**Specific Comments:**

*For the record: here are the 4 major issues that I think need to be addressed before publication: 1. Limited scientific significance. The paper does not really represent a major contribution to scientific knowledge. At best, it represents a (moderately novel) application of UM cascades to the simulation of conditional rainfall time series. The authors have a tendency to overstate the importance of the work. 2. Low writing quality. The English is neither fluent or precise. The text is difficult to read. There are a lot of long, complicated sentences. A major effort needs to be done in terms of writing. 3. The abstract needs to be completely rewritten to clearly explain what this paper is about. It should be more concise and clearly mention the main results. 4. There should be more details about the implementation of the method to make sure people can replicate the results. Ideally, example codes and/or datasets for testing ad comparing the method should be provided. It is not sufficient to refer to previous publications. Everything should be findable, accessible, interoperable, and reusable (FAIR). I have many more comments. But as I said above, I think it's not worth mentioning them here since the authors and I clearly disagree on the major issues.*

**Reply:** Comments 1,3 and 4 are simply reiterations of the referee's earlier remarks which we had already responded to. Unfortunately, the referee hasn't given any scientific reason why those replies weren't convincing enough, therefore we see no point in putting forth the same arguments again. Comment 2 on the other hand is a relatively minor one, and following the referee's suggestion we have simplified lengthy sentences in the revised manuscript.

**Response to Editor's Comments:**

**General comments:**

*As I read again the revised manuscript, I must agree with the general reviewers' comments that it is still too long, poorly written, and difficult to follow. "Scientifically", I believe the manuscript is fine (though not very novel) but I'm not convinced a non-expert in UM (most HESS readers) would understand and be interested in reading it in its current form. The following are some minor suggestions from my side, but I think you should revise the text considerably to improve its readability beyond my comments (see also the last reviewer's comments). I leave this to your judgment, but perhaps it would be helpful to ask a non-expert colleague for their thoughts on enhancing clarity. Please upload a revised version of the manuscript, which will be evaluated again by myself (it will not go to external review).*

**Reply:** We thank the editor for reading our manuscript, his positive remarks, his decision for a minor revision and providing suggestions for it.  We respectfully disagree with the editor's remark on the paper's novelty, since we are unaware of earlier studies that have proposed similar procedures to simulate realistic reference rainfall scenarios. Some parts of the revised manuscript have been rewritten to make the text a bit easier and more interesting for HESS readers.

**Specific suggestions:**

1. *The abstract is too long and should be more concise.*

   **Reply:** We make it as concise as possible in the revised version, but long enough to clearly point out the paper's novelty. We feel this is indispensable given the fact that referee #3 and the editor (as he mentions in his recent general comment) have often been rather doubtful regarding this.

2. *There are many repetitions in the text that can be removed (e.g., lines 107-108).*

   **Reply:**  Thank you for this comment, which we are taking care of.

3. *Line 153. TM is mentioned for the first time. Please check other abbreviations for the same issue.*

   **Reply:** Thank you again, same answer.

4. *Appendices should be presented as supplementary information, in a separate file.*

   **Reply:** Thank you again, same answer.

5. *Table 2 can be presented in the supplementary material. Also, the table is unclear - these are thresholds for what type of design?*

   **Reply:** We respectfully disagree. This table is important as it provides the governmental guidelines for reference rainfall obtained from regional rainfall zoning documents. Buildings/plots irrespective of the design or type of their storm-water management infrastructure are required to comply with certain drainage/discharge rules during the occurrence of such reference rainfall events.

6. *Figure 1 - consider also moving to the SI.*

   **Reply:** We feel it would be more informative near the introduction.

7. *Figure 5 and in the text. I would be more specific mentioning that the simulations are all temporal and not spatial.*

   **Reply:** Thank you for this comment, which we are taking care of.

8. *It will be beneficial for the readers to see an example of the analysis/model on GitHub or another repository as an example. In the text, you can refer to it, and simplify some of the explanations.*

   **Reply:** We partly agree with the referee, and are currently developing a consolidated Python library (MultiFractal Python Library) on GitHUb that will be dedicated to doing this and much more. However, we don't feel the need to focus on such software related issues

within the current scientific paper that is focused on the methodology, which is quite novel.